# Analysis of the Variability in Soil Moisture Measurements by Capacitance Sensors in a Drip-Irrigated Orchard

**DOI:** 10.3390/s20185100

**Published:** 2020-09-07

**Authors:** Jesús María Domínguez-Niño, Jordi Oliver-Manera, Gerard Arbat, Joan Girona, Jaume Casadesús

**Affiliations:** 1Efficient Use of Water in Agriculture Program, Institute of Agrifood Research and Technology (IRTA), Parc de Gardeny (PCiTAL), Fruitcentre, 25003 Lleida, Spain; jordi.oliver@irta.cat (J.O.-M.); joan.girona@irta.cat (J.G.); jaume.casadesus@irta.cat (J.C.); 2Department of Chemical and Agricultural Engineering and Technology, University of Girona, Campus Montilivi s/n, 17071 Girona, Spain; gerard.arbat@udg.edu

**Keywords:** capacitance sensor, HYDRUS-3D, soil water content, soil wetting patterns, soil temperature, two-steps calibration

## Abstract

Among the diverse techniques for monitoring soil moisture, capacitance-type soil moisture sensors are popular because of their low cost, low maintenance requirements, and acceptable performance. However, although in laboratory conditions the accuracy of these sensors is good, when installed in the field they tend to show large sensor-to-sensor differences, especially under drip irrigation. It makes difficult to decide in which positions the sensors are installed and the interpretation of the recorded data. The aim of this paper is to study the variability involved in the measurement of soil moisture by capacitance sensors in a drip-irrigated orchard and, using this information, find ways to optimize their usage to manage irrigation. For this purpose, the study examines the uncertainties in the measurement process plus the natural variability in the actual soil water dynamics. Measurements were collected by 57 sensors, located at 10 combinations of depth and position relative to the dripper. Our results showed large sensor-to-sensor differences, even when installed at equivalent depth and coordinates relative to the drippers. In contrast, differences among virtual sensors simulated using a HYDRUS-3D model at those soil locations were one order of magnitude smaller. Our results highlight, as a possible cause for the sensor-to-sensor differences in the measurements by capacitance sensors, the natural variability in size, shape, and centering of the wet area below the drippers, combined with the sharply defined variation in water content at the soil scale perceived by the sensors.

## 1. Introduction

The increase in the world’s population and the consequent increase in water consumption necessitates the proper and efficient management of water resources [1]. In agriculture, drip irrigation is one of the most important solutions for the efficient use of limited water resources [2]. Drip irrigation provides water to a limited volume of soil in the region where the greatest water extraction by plants occurs, reducing losses by surface evaporation and deep percolation [3,4,5]. The distribution of moisture within a volume of wet soil is known as the wet bulb [6]. Its formation is affected by a number of factors, including the physical properties of the soil (texture, bulk density, initial water content, etc.), absorption of the crops by the root system, soil surface evaporation, and the intensity of the irrigation rate [7,8]. 

Real-time monitoring of soil moisture can provide useful information for optimizing the amount and timing of irrigation [9,10]. Soil moisture can be measured using electromagnetic methods, such as time domain reflectometry (TDR) [11] and capacitance sensors [12], or using electrical resistance blocks [13], neutron probes [14], or tensiometers [15]. Among the range of different soil water sensing technologies, capacitance-type soil moisture sensors [16,17] are the most popular because of their cost, reasonable robustness and precision, low power consumption, and low maintenance requirements [18,19,20,21]. While their adoption by farmers remains low, their use is increasing not only for the visual supervision of the dynamics of soil water content (SWC) with irrigation, but also as a potential way of providing input data for decision support systems (DSS) that help to determine when to irrigate and how much water to apply on a plot [22,23]. 

However, although these sensors give good accuracy in laboratory conditions [17,19,24], in field conditions they show large sensor-to-sensor variability, especially in drip irrigation [9]. One explanation for this variability is that they only perceive a small volume of soil (in the order of 1 dm^3^ [25,26]), which makes them very sensitive to local variations in, for example, gravel content, bulk density, soil salinity, the existence of macropores and shrinkage cracks, the proximity of plant roots, and small-scale surface features [27,28]. Some of these factors vary little over time and, hence, once a sensor is installed their effect is permanent. In addition, there are other factors that influence sensor variability which are dynamic in nature. These include soil temperature [29] and soil apparent electrical conductivity [30]. The SWC pattern around a dripper is also dynamic and its change over time can be simulated with mathematical models. In this respect, HYDRUS [31] is a well-known software package for the simulation of water and solute movement in soils of one-, two- or three-dimensions, for different combinations of initial and boundary conditions [32,33]. Some authors [34,35,36,37] have used HYDRUS modelling to simulate surface drip irrigation. These and other works have demonstrated the ability of HYDRUS to simulate the space-time dynamics of soil water in drip irrigated crops. Simulations with HYDRUS can indicate whether the observed differences between sensors can be attributed to the expected dynamics of the wet bulbs or, alternatively, whether other factors need to be considered.

Despite the difficulties described above, capacitance-type soil moisture sensors are successfully being used for irrigation management, including in scenarios of drip-irrigated orchards [38,39]. A better understanding of the uncertainty of these measurements and of the variability in the actual soil conditions should provide clues about how to improve their performance, their effectivity and, ultimately, their practical utility in real orchard conditions.

The objective of this study is to analyze why capacitive sensors provide accurate measurements of SWC in laboratory conditions but, when installed in a real drip-irrigated orchard, show large sensor-to-sensor differences. The study consisted of analyzing the performance, during two consecutive irrigation seasons, of 57 capacitive-type soil moisture sensors installed in 10 soil locations around the drippers of an apple orchard under semi-arid conditions. The effect was considered of potentially disturbing factors such as variability in the soil area wetted by the dripper and the influence of soil temperature and sensor calibration. In addition, virtual sensor readings, obtained from simulations of soil water dynamics with HYDRUS-3D, were used to assess how much of the observed differences can be explained by the expected soil moisture dynamics of an idealized wet bulb.

## 2. Materials and Methods

### 2.1. Experimental Orchard

The research was carried out in the irrigation seasons of 2017 and 2018 in two experimental plots (Plot I and Plot II) of an apple orchard (*Malus domestica* Borkh. cv ‘Golden Reinders’) planted in 2011 and grafted on M-9 rootstock located at the IRTA-Lleida Experimental Station (Mollerussa, Lleida, Spain). The planting pattern was 3.50 m × 1.63 m with a north-to-south tree row orientation. The climate of the area is Mediterranean, with annual rainfall and evapotranspiration rates of 290 mm and 1093 mm, respectively, for the year 2017, and 506 mm and 1040 mm, respectively, for the year 2018. 

Irrigation water was supplied by a drip irrigation system (3.5 L h^−1^) with a 0.6 m separation between drippers. During the irrigation season, the water was applied at 8:00 a.m., except in the months of August and September 2017 when variations in irrigation schedules were applied. In general, these plots were irrigated, on a daily basis, with a daily irrigation dose (DID) to meet crop water needs based on the FAO water balance [40] (Equation (1)).
(1)$${\mathrm{DID}\;=\;\mathrm{ET}}_{o}{\;\text{×}\;K}_{c}$$
where ET_O_ is the reference evapotranspiration from the previous week, recorded by a weather station located on the same farm, and Kc is the crop coefficient determined in previous years in the same orchard using the weighing lysimeter method [41]. However, arbitrary irrigation doses were imposed in certain periods in order to test sensor response to soil water input/output imbalances. Furthermore, each plot was irrigated independently and, therefore, different doses of water were applied. Electrical conductivity of the irrigation water was 0.309 dS m^−1^, and NPK fertigation was applied (100 kg N, 30 kg P_2_O_5_ and 108 K_2_O) from May to June (90%) and in September (10%). 

The soil of the orchard was classified as *Typic Calcixerepts, coarse-loamy, mixed, and thermic* according to the Soil Survey Staff classification [42]. Soil samples were taken at different depths and their texture, bulk density, and organic matter were determined. The results obtained are shown in Table 1.

### 2.2. Soil Water Content Measurements

The soil moisture sensors used in this study, EC-5 and 10HS, are capacitance and frequency domain reflectometry (FDR)-type sensors (Meter Group Inc., Pullman, WA, USA). These sensors measure the dielectric constant or permittivity of the soil to calculate its moisture content. The EC-5 sensor is 5 cm long and has an approximate theoretical measurement volume of 0.3 L [21]. The 10HS sensor is 10 cm long and can measure 1 L of soil volume [20]. Both sensors measured the SWC every 10 s, and the average of 5 min was stored in the dataloggers CR800 and CR1000 (Campbell Scientific Inc., Logan, UT, USA), which used a multiplexer AM16/32 to increase the number of channels. The moisture sensors were deployed in two plots, with one sensor type in each plot. Three repetitions of 9 or 10 sensors were installed in each plot in different positions and depths around the dripper (Figure 1). The experiment ended in September 2018 when the sensors were removed from the soil and taken to the laboratory for two-step calibration under specific conditions described in a previous study [43].

The EC-5 sensors of Plot I were installed in 2013 and deployed in positions A, B and C at 15, 30 and 60 cm depths, with 3 repetitions each around a different dripper. All repetitions were within the same tree row and separated by a distance of less than 5 m. The 10HS sensors of Plot II, which were installed in 2016 and deployed in equivalent positions and depths as in Plot I, additionally included position D at a depth of 30 cm (Table 2).

A total of nine temperature probes (Omega HSTH-44000, 2252 Ohm) were also installed in 2013 in Plot I, at soil locations equivalent to those of the EC-5 sensors. Probe readings were recorded using the same datalogger as for the EC-5 sensors, through a dedicated multiplexer AM16/32 (Campbell Scientific Inc., Logan, UT, USA). Soil temperature measurements ended in early 2016 when the multiplexer was damaged by a flood. The dataset of soil temperatures analyzed in this work corresponds to the 2015 season, when soil temperature and soil moisture were recorded simultaneously. Figure 2 shows a temporal scheme that indicates the most relevant moments related to the installation of soil moisture and temperature sensors, calibration, and simulation with HYDRUS-3D.

The extent and position of the area wetted by drippers was characterized in this site in order to standardize the positions of the sensor in relationship to the wet bulbs. The measurements were done at the end of an irrigation pulse, in July–August 2018, using a portable Fieldscout TDR 300 soil moisture instrument (Spectrum Technologies INC., Aurora, IL, USA) with 12 cm-long rods (Figure 3). The characterization of the extent of the wetted area consisted in measuring the SWC at intervals of 10 cm, parallel and perpendicular to the dripline. The position of the wetting pattern was referred to the centering of the wet bulb relative to the dripper and was determined as the point with the highest SWC between two drippers. To determine the variability in the extent and position of the wetted area, a “reference wetted pattern” was defined as the wetting pattern most frequently observed during the measurements. Then, all transects included in the dataset were compared with this “reference wetting pattern”.

SWC at Plot I was measured periodically using a neutron probe Twelve access tubes were installed in 2013 at positions A, B, C, D, relative to the drippers, and repeated in three drippers as described in Domínguez-Niño et al., 2020 [32]. The volumetric soil water content in these access tubes was measured using a neutron probe (Hydroprobe 503DR, Campbell Pacific Nuclear Corp., Martinez, CA, USA) which had previously been calibrated for this site. Measurements were taken at depths between 0.20 m and 1.00 mat intervals of 20 cm depth, on a total of 15 days in the periods from May to October of 2017 and 2018.

### 2.3. HYDRUS-3D Model

The HYDRUS-3D model (v. 2.02) was used to simulate the soil water movement in a three-dimensional domain and hourly scale [31]. The water movement was simulated for 2017 and 2018. The Richards equation (Equation (2)) governs the movement of water flow in an unsaturated soil, with a sink term, S (cm^3^ cm^−3^ day^−1^), incorporated to consider water absorption by the root system.
(2)$$\frac{\partial \theta }{\partial t}\;=\;\frac{\partial }{\partial x}\left[K\left(h\right)\frac{\partial h}{\partial x}\right]\;+\;\frac{\partial }{\partial y}\left[K\left(h\right)\frac{\partial h}{\partial y}\right]\;+\;\frac{\partial }{\partial z}\left[K\left(h\right)\left(\frac{\partial h}{\partial z\;}+\;1\right)\right]-\;S$$
where *θ* represents the volumetric water content (cm^3^ cm^−3^), *h* is soil water pressure head (cm), *t* is time (days), *x* and *y* are the horizontal space coordinates (cm), *z* is the vertical space coordinate (cm), and *K* is hydraulic conductivity (cm day^−1^). In HYDRUS, the *S* term represents the volume of water extracted by the roots in a soil volume unit per unit of time. It uses a complex function proposed by Feddes [44] (Equation (3)):(3)$$S\left(h,z\right)\;=\;\alpha \left(h\right)S_{max}\left(h,z\right)$$
where α is a dimensionless water stress reduction factor expressed as a function of pressure head h (cm), whose values were taken from Taylor and Ashcroft [45] for deciduous fruit trees. *S_max_* (cm^3^ cm^−3^ day^−1^) is the maximum possible root water extraction rate when soil water is not a limiting factor, and z is the soil depth (cm).

The HYDRUS model solves the Richards equation using van Genuchten’s parametric function [46], which relates moisture and soil water potential through Equation (4):(4)$$\theta \left(h\right)=\;\{\begin{array}{c} \theta_{r}\;+\;\frac{\theta_{s}\;-\;\theta_{r}}{{\left[1\;+\;{\left|\alpha \cdot h\right|}^{n}\right]}^{m}}\;h\;<\;0\\ \theta_{s}\;h\;\geq \;0 \end{array}$$
where *θ_s_* (cm^3^ cm^−3^) is saturated water content, *θ_r_* (cm^3^ cm^−3^) is residual water content, and *m*, *n*, and α are empirical values that affect the shape of the retention curve (for purposes of simplification it is assumed that *m* = 1 – (1/*n*)). Unsaturated hydraulic conductivity, *K*(*h*) (cm day^−1^), is determined through Equation (5) [47]:(5)$$K\left(h\right)=\;K_{s}S_{e}^{l}{\left[1\;-\;{\left(1\;-\;S_{e}^{1/m}\right)}^{m}\right]}^{2}$$
where *S_e_* is the dimensionless effective water content, *K_s_* is the saturated hydraulic conductivity of the soil and *l* is an empirical parameter related to the conductivity between the pores.

The soil hydraulic parameters and root distribution used for the purposes of the present study were obtained from a previous work [38] in which the most appropriate HYDRUS-3D configuration was defined and the soil water dynamics in a drip irrigated orchard were simulated. However, in contrast with the previous work, here the initial conditions were established from the SWC measured by the capacitance sensors at the beginning of the year. We also assumed a semi-circular area with a radius of 10 cm, which was the waterlogged area during the irrigation. Accordingly, the flux, q (Equation (6)), was estimated as:(6)$$q\;=\;\frac{{\mathrm{Emitter}\;\mathrm{discharge}\;\mathrm{flow}\;\mathrm{rate}\;(\mathrm{cm}}^{3}{\;h}^{-1})}{{\mathrm{wetted}\;\mathrm{surface}\;\mathrm{area}\;(\mathrm{cm}}^{2})}\;=\;\frac{{3500\;\mathrm{cm}}^{3}h^{-1}}{{157\;\mathrm{cm}}^{2}}\;=\;22{.29\;\mathrm{cm}\;h}^{-1}$$

Virtual sensors were defined within the simulated geometry in order to monitor the soil water dynamics from equivalent locations to those where the capacitance-type soil moisture sensors had been installed in the field. Three virtual sensors were defined for each of the 10 soil locations of interest, one centered at the position of interest and the other two displaced 10 cm closer to and further from the dripper, respectively.

HYDRUS-3D model is characterized by simulating soil water dynamics in a homogeneous soil and root distribution where ideal, symmetric, and centered wet bulbs develop around the dripper. However, HYDRUS-3D neither represent heterogeneous soil and root distributions, or macropores and soil irregularities among other phenomena that usually take place in a real soil where wet bulbs are generated.

### 2.4. Analysis of Sensor Performance

Sensor performance was analyzed using quantitative indicators for several aspects of interest, which were calculated as follows:-**Repeatability between sensors:** refers to the variability between sensors installed at equivalent depth and position relative to the dripper. They were quantified as the root mean square error (RMSE) between those repetitions, using the dataset composed of the daily values of Plot I and Plot II in 2017 and 2018. In the case of the HYDRUS-3D simulations, the repetitions came from the 3 virtual sensors defined for each of the 10 locations of interest.-**Sensitivity to the soil water balance:** refers to the dependence of the SWC at a given sensor location on the balance of water inputs/outputs to the soil. This indicator was quantified through a regression that modelled the sensor measurement of any given day as a function of the sensor measurement 7 days earlier, the balance that day, and the aggregated balance of the previous 7 days.
(7)$${\mathrm{SWCd}}_{d}\;=\;Coef_{0}{\text{·}\mathrm{SWCd}}_{d-7}\;+Coef_{1}{\;\text{·}\mathrm{bal}}_{d}\;+\;Coef_{2}{\text{·}\Sigma \mathrm{bal}}_{d-7\text{…}d}$$
where:-SWCd_d_: the driest SWC measured by the sensor on day d, cm^3^ cm^−3^.-SWCd_d−7_: the driest SWC measured by the sensor 7 days earlier (d − 7), cm^3^ cm^−3^.-bal_d_: the balance of water inputs and outputs (DID_d_ + PPT_d_ – ET_d_), mm.-DID_d_: the daily irrigation dose on day d, mm.-PPT_d_: the daily rainfall dose on day d, mm.-ET_d_: the daily irrigation dose on day d, mm.-Σbal_d-7...d_: the aggregated balance of water inputs and outputs in the previous 7 days (Σ(DID_d_ + PPT_d_ – ET_d_)), mm.-*Coef*_0_, *Coef*_1_, and *Coef*_2_: the regression coefficients.

To focus on water balance variations related with irrigation, the analysis included only days in the irrigation season and excluded rainy days and the day following rain. This linear regression model was analyzed using the Python package statsmodels [48].

### 2.5. Statistical Calculations

In order to facilitate their comparison, in this study both the uncertainties in the measurement process and the variability of the measured data were expressed in terms of the root mean square error (RMSE). In accordance with their usage in diverse disciplines [49,50,51,52,53], here, uncertainty refers to the degree of precision with which a quantity is measured, while variability refers to the natural variation in some quantity. We use the term variability to refer to the measured data and, also, to refer to the presumed real quantity, since we have no way to distinguish between them.

The coefficient of determination (R^2^) and the RMSE which were used for the statistical analysis were calculated as follows.

The R^2^ (Equation (8)) value explains how much of the variability of a factor can be caused or explained by its relationship to another factor. It is computed as a value between 0 and 1. Values close to 1 indicate a good agreement of the model.
(8)$$R^{2}\;=\;\frac{{\left[\sum_{i=1}^{N}\left(O_{i}\;-\;\bar{O}\right)\left(S_{i}\;-\;\bar{S}\right)\right]}^{2}}{\sum_{i=1}^{N}{\left(O_{i}\;-\;\bar{O}\right)}^{2}\cdot \;\sum_{i=1}^{N}{\left(S_{i}\;-\bar{S}\right)}^{2}}$$

The RMSE (Equation (9)) measures how much error there is between two data sets. It compares a predicted value and an observed value. Values close to 0 indicate a better fit of the model.
(9)$$\mathrm{RMSE}\;=\;\sqrt{\frac{\sum_{i=1}^{N}{\left(O_{i}\;-\;S_{i}\right)}^{2}}{N}}$$
where *N* refers to the number of compared values, *O*_i_ the ith observation point, *S_i_* the *i*th simulation, and *Ō* the observed mean value.

## 3. Results 

### 3.1. Variability in the Soil Conditions around a Dripper

#### 3.1.1. Centering and Extent of the Wetted Area

The extent of the soil surface wetted by the drippers and the alignment of the centers of these areas with the drippers were studied in order to obtain a clue to the variability in a real orchard of the size, shape and centering of the wet bulbs. The measured SWC transects around drippers, following the dripline and perpendicular to it, are illustrated in Figure 4. The width of the wetted area following the dripline was 64.5 ± 8.9 cm, which partly overlapped with the area wetted by the neighboring dripper. In the axis perpendicular to the dripline, the width was 87.6 ± 13.7 cm. The centering of the wet bulb was displaced westwards by 13.6 ± 7.5 cm from the emitter, towards the center of the tree line. 

Compared with this range of variabilities in the wetted area, the separation between sensor positions A–B and A–C was 30 cm. The volume of sensitivity of sensor EC-5 is around 0.3 L, which corresponds, approximately, to a horizontal cylinder of 5 cm diameter and 9 cm length. For sensor 10HS the volume of sensitivity is around 1 L, which corresponds, approximately, to a horizontal cylinder of 9 cm diameter and 21 cm length. These volumes of sensitivity suggest a fine spatial resolution which, for positions B and C, may fall in a soil region with variable inclusion within the wetted area. Based on the measurements of these wetted areas, the RMSE values of the SWC at positions A, B, and C were 0.039, 0.075, and 0.095 cm^3^ cm^−3^, respectively.

#### 3.1.2. Pattern of Temperature in a Soil Wet Bulb

Soil temperature at the locations where the studied moisture sensors were located varied following both a seasonal pattern and a daily pattern. Soil temperature measurements ended accidentally in early 2016. We considered that the simultaneous recording of soil temperature and soil moisture by EC-5 probes in 2015 was sufficient to assess the magnitude of the effect of temperature on the measurements of soil moisture and that this magnitude was also representative for 2017 and 2018. Over the irrigation season, the daily mean soil temperature ranged between 19 and 27 °C, with instantaneous maximum and minimum values of 16 and 34 °C, respectively. The amplitude of the daily pattern varied with position and depth, with location C15 showing the widest daily amplitude of up to 6 °C (Figure 5). Weather conditions were observed to affect temperature, with rainfall producing sudden drops in temperature at all positions and depths, of as much as 8 °C, and progressive recovery in the following days. 

At the sensor locations with the widest fluctuations in soil temperature, the SWC readings were checked for signs of temperature effects (Figure 6). EC-5 sensors at C15 showed a daily pattern with minimum and maximum values synchronized with minimum and maximum values of soil temperature. The relationship between measurements by those sensors during the night and simultaneous measurements of soil temperature showed a significant slope of up to 0.002 cm^3^ cm^−3^ per °C. On the other hand, measurements by 10HS sensors did not show such clear signs of being influenced by temperature, though some of the sensors recorded a nocturnal decrease in SWC, parallel to the decrease in soil temperature, with a slope that in all cases was lower than 0.001 cm^3^ cm^−3^ per °C. Considering a range of fluctuation in soil temperature of up to 8 °C, caused either by weather or the diurnal cycle, a high estimate of its potential impact on SWC measurements would be a deviation smaller than 0.016 and 0.008 cm^3^ cm^−3^ for EC-5 and 10HS, respectively. This high estimate would correspond to position C15, with lower potential impacts in other sensor locations.

### 3.2. Overall Response of the Sensors

The SWC data were recorded from 57 moisture sensors deployed in two plots during 20 consecutive months in 2017 and 2018. After this period, the sensors were recovered and characterized in the laboratory [43]. Different patterns of sensor response were observed, both over the course of a day and at a seasonal scale, among the depth and positions where the sensors were located. Generally, over the course of a day (Figure 7), in position A for each of the depths, the sensors were the most sensitive to irrigation and responded quickly to the irrigation cycles as well as to the lack of irrigation. In this location, the sensors showed high sensor-to-sensor differences and a wide daily amplitude of SWC between the minimum before irrigation and the maximum following irrigation. In position B, the sensors followed a similar pattern to that of position A, but their response to irrigation was more delayed in time. The sensors installed in this position tended to show a smaller amplitude between SWC before and after irrigation, especially at the depth of 60 cm. In position C, some of the installed sensors followed a seasonal pattern with only a faint effect of the irrigation cycles. In position D, the sensors followed a seasonal pattern related with occasional rains and overall drying in the periods of high ETo, with no perceptible effects of the irrigation cycles. 

The amplitude of the daily oscillation of SWC between its minimum before an irrigation cycle and its maximum after an irrigation cycle was calculated both in capacitance sensors and in HYDRUS-3D during and outside the irrigation season (Figure 8). 

According to the results, during the irrigation season the capacitance sensors at soil location A15 showed a wide oscillation during the irrigation cycle, with a median of 0.027 cm^3^ cm^−3^ and an 80th percentile as high as 0.069 cm^3^ cm^−3^. These amplitudes were approximately halved at A30, B15, and B30 and were much further reduced at 60 cm depth. In position C, the median amplitude was in the order of 0.004 cm^3^ cm^−3^ or smaller but with a large variability, as shown by an 80th percentile of up to 0.029 cm^3^ cm^−3^ in C30. All amplitudes were much smaller outside the irrigation season, with all medians below 0.003 cm^3^ cm^−3^ and the 80th percentiles at 0.01 cm^3^ cm^−3^ or smaller.

For their part, the HYDRUS-3D simulations showed in Figure 8 a similar order of amplitude to that of the sensors for locations A30, A60, B30. However, the median amplitude at A15 simulated by HYDRUS-3D was only 0.0089 cm^3^ cm^−3^, one third of that observed by sensors, and also with less variability, with an 80th percentile of 0.0143 cm^3^ cm^−3^. The wider oscillation with sensors at A15 consisted of a more intense drop of SWC before irrigation compared with simulation. At this position, after irrigation and redistribution, the simulation measurements tended to decrease less with water uptake by roots. At C15 and C30, the median amplitude simulated by HYDRUS-3D was three times higher than that observed by the sensors, but with less variability. Overall, at C positions most sensors only showed a faint oscillation with irrigation cycles, whereas the oscillations produced by irrigation were clearer and more intense in the simulations. Hence, the soil water dynamics reproduced in the HYDRUS-3D simulations may explain the patterns of daily amplitude observed in sensors at A30, A60 and B30, but do not explain either the large amplitude observed at A15 or the much more attenuated C position amplitudes. In this respect, it seems that the simulations considered a more even distribution of soil water and also of water uptake by roots among the different positions. Compared to the sensors, the simulations were less variable, underestimated uptake at A15, and overestimated the influx of irrigation water at positions C. 

The timing of irrigation affected the SWC pattern over the course of a day. Figure 9 illustrates the dynamics of SWC when irrigation was in the morning, split between morning and afternoon, and in the afternoon, showing the sensor-recorded data at different locations in the soil together with the corresponding HYDRUS-3D simulation in the 2017 season. At position A15, irrigation in the morning produced a steep drop of SWC shortly after irrigation, with SWC during the night at the lower end of the daily range. At the other extreme, irrigation in the afternoon resulted in an attenuated drop after irrigation, with SWC remaining high, near field capacity, overnight, and dropping the following day before irrigation. With irrigation split in two pulses per day, SWC remained high for a longer period per day and, in this particular case, dropped to its daily minimum before the second pulse. The effect of the timing of irrigation on the SWC value at night justifies usage of the minimum daily SWC as a summary of the daily cycle, rather than the average daily SWC which would be much more affected by the timing of irrigation. The daily patterns described by the HYDRUS-3D simulations agreed with those described by the sensors at A15 in the rise of SWC during irrigation but differed in that, following irrigation, the relaxation of SWC was smoother and without a clear drop at the hours of higher ET. The patterns at other sensor locations in the soil were more attenuated both when measured by sensors and when simulated.

### 3.3. Variability between Sensors at Seasonal Scale

#### Repeatability between Sensors

Sensors installed at the same depth and position relative to the dripper tended to show synchronized patterns in terms of their response to irrigation cycles. However, the series of SWC measurements tended to fluctuate within a particular range for each individual sensor. In order to quantify sensor-to-sensor differences, we calculated the RMSE between measurements taken at the same time by sensors installed in equivalent locations with respect to depth and position relative to the dripper. This indicator was calculated from the daily values t including both plots and both irrigation seasons—i.e., excluding the periods of the year without irrigation. The results are shown in Figure 10, together with the equivalent indicators calculated from the HYDRUS-3D simulations. Overall, sensor-to-sensor repeatability depended on the location where the sensors were installed, ranging between 0.020 cm^3^ cm^−3^ and 0.050 cm^3^ cm^−3^. In general, the closer to the dripper, either in position or in depth, the higher the RMSE. Sensors located in position A had the lowest repeatability, in particular at the depth of 15 cm (0.050 cm^3^ cm^−3^), while repeatability improved at the depth of 60 cm (0.035 cm^3^ cm^−3^). Sensors located in position B showed greater repeatability than sensors located in position A. In particular, the depths of 15 cm and 30 cm showed greater repeatability (0.027 cm^3^ cm^−3^ and 0.037 cm^3^ cm^−3^) than the depth of 60 cm (0.022 cm^3^ cm^−3^). In position C, sensors located at the depth of 15 cm had lower repeatability (0.047 cm^3^ cm^−3^) than the rest of the depths, while the sensors located at the depths of 30 and 60 cm showed greater repeatability than sensors located at the depth of 15 cm (0.024 cm^3^ cm^−3^ and 0.021 cm^3^ cm^−3^, respectively). 

In contrast with sensors, the RMSE between simulations where the horizontal position of the virtual sensor was displaced 10 cm to either side, was always below 0.020 cm^3^ cm^−3^ in all locations and below 0.006 cm^3^ cm^−3^ in A and B positions at all depths. This would suggest that variability in the alignment between wet bulb and dripper, alone, cannot explain the observed variability between sensors. In addition, in position C, the simulated RMSE results for the depths of 30 and 60 cm were close to the RMSE of the sensors, with values of 0.016 cm^3^ cm^−3^ and 0.018 cm^3^ cm^−3^, respectively. 

### 3.4. Seasonal Pattern at Each Sensor Location in the Soil

Over the course of the two monitored irrigation seasons, the data recorded by the capacitance soil moisture sensors displayed different seasonal patterns of response to irrigation depending on the location of the sensor in the soil. Figure 11 and Figure 12 show the seasonal soil water dynamics at the different sensor locations measured with capacitance sensors and simulated HYDRUS-3D model, irrigation, rainfall, and ET measured by the weighing lysimeter in the years 2017 and 2018. The observed seasonal patterns differed between the two monitored plots. In general, although more evident in Plot I, sensor data from different positions and depths tended to be closer after rain and in periods of over-irrigation and more separated in periods with drier conditions. In periods of higher water deficit, variability between sensor locations increased. In Plot I under this situation, both the HYDRUS-3D and neutron probe SWC measurements were higher than those from the capacitance sensors. 

During the irrigation season, irrigation cycles affected capacitance sensor measurements in differing ways depending on sensor location. Compared with measurements by neutron probe, EC-5 sensor measurements in Plot I underestimated SWC in all positions and depths. Despite this bias, irrigation cycles as well as the lack of irrigation produced an immediate response in sensor measurements, especially in positions A. In contrast, in the other positions the SWC decreased more slightly over time. At the depth of 30 cm, the seasonal variability was less pronounced. The positions closest to the dripper (position A) tended to respond quickly to irrigation cycles, evapotranspiration, and consumption by the crop on the same day. The response at positions B and C tended to be slower and more progressive. At the depth of 60 cm, especially in positions away from the dripper, the sensors recorded less variation in response to the irrigation cycles. In position D, the sensors showed a slowly varying trend throughout the season, except for some peaks associated with rainfall.

### 3.5. Comparisons between Capacitance Sensor Measurements and HYDRUS-3D Simulations

All the results reported thus far correspond to the use of capacitance-type sensors with their factory calibration for mineral soils, which is their most expected usage in commercial farms. In order to determine whether the results would improve with a calibration specific for the soil at that orchard, a previously developed two-steps calibration [43] was performed. Overall, the R^2^ varied between sensor positions and depths regardless of the calibration used. In general, higher R^2^ were observed in the sensor locations that were deepest and farthest from the dripper. In general, the RMSE between measurements by capacitance sensors and neutron probe decreased with the two-step calibration, except for the depth of 15 cm, where it increased.

Figure 13 shows the effect of calibration on the fit between measurements by 10HS sensors and simulations by HYDRUS-3D. The data shown corresponds to one measurement every 15 days for the whole studied period—both during and outside the irrigation season—during 2017 and 2018. In general, with factory calibration the sensor-measured SWC was smaller than that of the simulations, except in the case of a few A, B and C positions at the depth of 15–30 cm. Overall, application of the soil-specific calibration did not reduce the scatter of the sensor measurements but did improve the adjustment of the whole cloud of sensor measurements to that of the simulations.

### 3.6. Sensor Sensitivity at Each Location to Fluctuations in the Balance of Water Inputs/Outputs 

A feature of special interest when comparing different sensor locations is how sensitive each location is to the balance between water inputs/outputs to the soil. This information can be used to highlight candidate sensor positions to monitor the fit between irrigation dosage and crop water requirements. In addition, visual interpretation of the sensors can indicate which locations may respond swiftly to the latest irrigation cycle and which may respond more progressively to the outcome of a period including several irrigation cycles.

The dependence of sensor-measured SWC on the balance of water input/outputs is summarized in Table 3. The analysis consisted of checking how the SWC measured on a given day could be modelled as a function of the SWC measured a week earlier, the water balance of that day, and the aggregated water balance of the previous week. The results show that this dependence varies according to the position and depth where the sensor is located. In all cases, the independent variable with the strongest effect was the measurement one week earlier, which highlights the relevance of the trends rather than the absolute values in the interpretation of SWC. As shown in the table, the response at positions A depended on the SWC measured the previous week and the water balance of the same day, especially at the depths of 15 cm and 30 cm. In this respect, sensors in position A responded as if they had no memory of the balance of the previous week. Meanwhile, the B positions depended on the SWC measured the previous week, the water balance of the same day (especially at depth of 30 cm) and the water balance of the whole week (especially at 30 cm). The C positions depended on the SWC measured the previous week, the water balance of the same day (in particular at the depths of 30 cm and 60 cm) and the water balance of the whole week (especially at depths of 15 cm). The D positions depended on the SWC measured the previous week, the water balance of the same day, and especially the water balance of the whole week.

Figure 14 illustrates an example of the distinct type of response to the soil water balance at different sensor locations. In this example, during several days in June 2018, in Plot II, the applied irrigation doses were reduced for experimental purposes. Measurements of SWC by capacitance sensors in the following days at locations A15, A30, B15, and B30 all revealed the occurrence of an irrigation deficit and, later on, the recovery of the soil moisture when irrigation doses were increased again. However, the recovery of SWC at position A when irrigation doses were increased again may take place almost the same day (for instance sensor A30 in this example), as if the sensor had no memory of the preceding period of deficit. On the other hand, sensors at position B tended to have a longer period of response, thus aggregating the outcome of several irrigation cycles. Sensors at positions C and D were also affected by the deficit but took a much longer period to recover. Meanwhile, sensors at 60 cm depth showed a much fainter response to these changes in irrigation.

The analysis of sensitivity of different sensor locations to the soil water balance was also performed with the HYDRUS-3D simulations and the results are summarized in Table 4. In the case of simulations, the SWC at all depths and positions depended on the SWC measured in the previous week and the water balance of that day (especially at the depth of 30 cm). Additionally, in positions A and B at the depth of 60 cm, there was a highly significant dependence of the SWC on the water balance of the previous week. Interestingly, at sensor locations A15, A30, B15, and B30 the balance of the previous week was not significant, thus suggesting that the lack of memory of the preceding period is a sound and repetitive feature at these locations, well represented in the soil water dynamics by HYDRUS-3D.

### 3.7. Components of the Variability in the Measurements by Capacitive-Type Soil Sensors

Regarding the performance of sensors in real orchard conditions, an issue of interest in this study was to compare the uncertainties in the measuring process with the observed variability in sensor data and with the natural variability in the soil environment. To this end, all them were expressed in the same terms, as RMSE. When comparing the output of several soil sensors, the observed differences between sensors can be caused by a combination of uncertainties in the measuring process and actual variability in the physical property being measured. In this study, the distinction between uncertainty in the process and variability in the data can offer clues in terms of directions for improvement in the usage of the sensors. However, in a practical application there may be no need to distinguish between them [48] and the whole ensemble would contribute to the overall uncertainty of monitoring SWC in a drip-irrigated orchard. That is to say, when using a setup of several sensors, each of them reporting a different value of SWC, usage of these data for decision-making is faced with the uncertainty resulting from the combination of the measuring process and the natural variability in the actual values. The preceding sections described the observed variability in sensor measurements at different locations in the soil around a dripper. In order to better appraise its significance and gain clues as to its possible origin, the observed variability can be compared with the uncertainties in different factors affecting the process of measuring SWC by capacitive-type sensors (Figure 15).

First, the accuracy of SWC measurement by capacitive-type sensors can be decomposed into two steps [18,54]. The first step converts the sensor response to permittivity, regardless of the media—where it has been measured. The second step converts permittivity to SWC for a specific soil. The 10HS sensors used were, at the end of the study, calibrated specifically for the soil of the orchard [43]. Comparing the output of the different options for sensor calibrations, the range of uncertainty for the first and second steps, expressed as RMSE, were 0.006 cm^3^ cm^−3^ and 0.014 cm^3^ cm^−3^, respectively. Here, the first step includes the variability between individual sensors and the second step the specific relationship between permittivity and SWC for that soil. In addition, soil temperature varies with depth and with position relative to tree shade and fluctuates over the course of a day, with potential effects on 10HS sensor measurements which we estimated as an uncertainty of up to 0.008 cm^3^ cm^−3^.

Virtual sensors that monitored HYDRUS-3D simulations at the sensor locations ±10 cm in the direction to the dripper showed sensor-to-sensor differences of between 0.003 cm^3^ cm^−3^ and 0.018 cm^3^ cm^−3^, depending on sensor location. In contrast, in this study, the observed differences between capacitive-type sensors in real drip-irrigated orchard conditions ranged between 0.021 cm^3^ cm^−3^ and 0.050 cm^3^ cm^−3^, depending on sensor location. In addition, the range of uncertainty between positions A, B, and C showed a range of differences between 0.048 cm^3^ cm^−3^ and 0.091 cm^3^ cm^−3^.

This comparative range of variability suggests the possibility of ranking the factors to consider for optimizing the monitoring of soil moisture with capacitive-type sensors in a drip-irrigated orchard. In particular, it highlights the uncertainty derived from the arbitrary movement of water at the soil surface, between the dripper and the entrance into the soil. In contrast, there seems to be little margin of improvement in direct sensor response, as two-step calibration improves measurement accuracy but is minimal compared to other sources of uncertainty unless it manages to integrate a much larger soil volume.

## 4. Discussion

### 4.1. Variability in the Soil Conditions

The performance of capacitance moisture sensors (EC-5 and 10HS) installed at different depths and positions relative to the dripper were evaluated for two years. In addition, the particular conditions of disturbance and the natural variability of soil water patterns in drip irrigation was evaluated. In this type of scenario, the actual environment around a sensor can vary considerably, combining modellable and capricious patterns, and might, at least partly, cause the observed variability in sensor measurements. To some degree, sensor-to-sensor differences can be expected, given the small volume of influence of capacitance sensors and the highly heterogenous patterns of SWC expected in their vicinity. Virtual sensors were configured in the HYDRUS-3D model to monitor the simulated SWC at the locations where the real sensors were installed and, to check sensitivity to the precise position of the sensor, additional virtual sensors were displaced 10 cm from their original position in either direction relative to the dripper. Overall, the virtual sensors were much more repetitive and with slighter differences between soil locations compared to the real sensors. Our characterization of the soil area wetted by the drippers show that the size, shape, and alignment of these areas have a large natural variability. Even at locations close to the vertical of the dripper (position A), measurements of SWC in the uppermost 12 cm show a large variability which suggests that the wet bulb is not always centered there. For locations presumably closer to the border of the wet bulb (positions B and C), SWC variability in the uppermost 12 cm was still larger. These measurements of the wetted area, close to the wet surface, are not a direct measurement of the wet bulb, which develops deeper. However, the observed variability in the wetted area may be indicative of the variability of the wet bulb, though as observed with sensors, up to a point, variability tends to diminish with depth.

Soil temperature in the wet bulb is also heterogeneous. First, it depends on depth, with shallower locations showing wider diurnal variations, with an amplitude in summer of up to 8 °C at 15 cm depth and 1 °C at 60 cm depth. Second, in our measurements it varied with the position relative to the dripper, with diurnal variation increasing with the distance from the dripline. This may be attributable to differences in shade [55]. Differences in soil moisture might also imply thermal differences, either through soil evaporation or through the thermal conductivity of the soil, which is dependent on soil moisture [56]. In our results, EC-5 sensors showed clear signs of being affected by temperature, with a potential disturbance up to ±0.02 cm^3^ cm^−3^. The effect of temperature on EC-5 sensors has been described by other authors [21]. There was less clear evidence of 10HS sensors being affected by temperature, though this cannot be discarded since temperature effects have been observed by other authors [57]. The dynamic behavior of the two variables, soil temperature and SWC, in the context of a drip-irrigated orchard where they vary diurnally and in space within the dripper frame, can disguise their relationship. Laboratory characterization of 10HS sensors by other authors has shown that temperature effects depend on soil texture, with an increase of the measured SWC with temperature in soils with fine texture [58]. Given the observed range of variability in soil temperature (between 1 and 25 °C through the year), the potential effect of soil temperature on the measurement of SWC at an annual scale can be as high as 0.028 cm^3^ cm^−3^ comparing different extreme conditions through the year. Additionally, in other moisture capacitance sensors such as the SMT100 model, it has been reported that temperature has a significant influence on sensor output [17]. As stated above, this effect would be higher at shallow and sunlit positions and lower at deeper and shaded positions. 

Regarding the diurnal pattern of SWC, we observed the widest amplitude at position A15, near the area where irrigation water enters the soil. The diurnal amplitude displayed by sensors at this soil location is much wider and more variable than expected according to HYDRUS simulations (Figure 7). This difference might be attributable to the more idealized and smooth conditions at the soil surface represented in the simulations. In real orchard conditions, the soil surface presents an arbitrary microrelief which determines preferential pathways at the soil surface to water from the dripper and may produce arbitrary patches of waterlogged soil during irrigation [59]. This coexists with patches of differently shaded/sunlit soil spots, which in turn may determine heterogeneous patterns of evaporation at the soil surface. All this would determine a more arbitrary and sharply defined soil condition scenario compared with that represented in the simulations. Even so, the simulations coincided with sensors in terms of the ranges of diurnal amplitude at 30 cm depth in positions A and B, a region in the soil which is at the core of the daily cycles of hydration and water uptake by roots. The simulations also estimated a greater amplitude than observed by sensors at 60 cm depth, where they might overestimate root water uptake compared with sensor measurements, and at position C, where they might overestimate the arrival of water from the irrigation pulses. The range of diurnal fluctuations is much smaller offseason, where, except for rain events, the magnitude of the common water inputs and outputs to the soil are much smaller and, additionally, the wet bulbs disappear. 

As expected, the patterns of diurnal fluctuation in SWC were affected by the timing of irrigation, particularly in relation to the daily curve of ET. Irrigation in the early morning produces steeper changes as the result of saturation during irrigation followed by redistribution and immediate uptake by roots. In contrast, when irrigation is in the afternoon, SWC remains high at night and drops the following day. Irrigation split into two pulses, one in the morning and the other in the afternoon produces a less severe pattern for a longer period of time. Given the effect of the timing of irrigation on SWC at night and this, in turn, on the daily average, the driest daily measurement appears as a practical and robust summary of the preceding cycle of irrigation, redistribution, and uptake by roots.

### 4.2. Repeatability between Sensors

The characterization of capacitance-type soil moisture sensors in laboratory conditions has produced highly repetitive readings among sensors [21,43]. Nevertheless, in field conditions, large differences in sensor measurements have been reported by many authors [26,60,61,62,63]. Our results show large sensor-to-sensor differences, even though repeated sensors were installed precisely at the same soil locations in terms of depth and position relative to the dripper. We quantified these differences as RMSE and observed that it depended on sensor location. Broadly speaking, the RMSE between equivalent sensors seems to vary with depth, from 0.05 cm^3^ cm^−3^ at 15 cm to 0.02 cm^3^ cm^−3^ at 60 cm depth. At 30 and 60 cm depth, variability decreased from position A to positions B and C. A correspondence can be suggested between the positions with the largest diurnal amplitude and their sensor-to-sensor differences. In this sense, some sensor positions in the soil are more sensitive to the irrigation cycles, and any factor that may affect either hydration or water uptake at those locations would greatly contribute to sensor-to-sensor differences. 

The HYDRUS-3D simulations were found to be much more repetitive, including between virtual sensors displaced 10 cm in either direction relative to the dripper. With the HYDRUS-3D simulations, an RMSE was calculated of around one order of magnitude smaller than that observed with the sensors in positions A and B, while it was similar to the sensor-based RMSE at position C at the 30 and 60 cm depths. That is, the simulations produced a more homogeneous SWC within the wet bulb than measured by sensors, and only in the periphery of the bulb were the simulations sensitive to the precise centering of the bulb.

In our measurements, even though we ensured the equivalent depth and position of the repeated sensors, our results in terms of variability in soil wetting patterns and the existence of heterogeneous SWC and soil temperature patterns suggest that the spatial coordinates of the sensor do not guarantee that they will encounter repeat conditions in terms of irrigation cycle dynamics. The reported 10HS sensor volume of sensitivity is around 1 L [19,25] and, furthermore, sensitivity is not homogeneous within this volume. Therefore, any soil property that may vary at this spatial scale would cause the immediate vicinity of a particular sensor to depart from the idealized properties considered in simulations. For instance, sensors would be sensitive to the presence of macropores and stones [64], microvariations in soil bulk density [65], uneven distribution of roots [66], uneven temperature due to contrasts between shaded and sunlit soil surface [67], etc. Moreover, there may be additional interaction between these factors. In contrast, HYDRUS-3D simulations do not allow the input of a dynamic root system [68] and do not consider that the soil structure may contain macro-pores or pores with less tortuosity and higher continuity [69]. In addition, the simulations could be improved by implementing the possibility of simulating the formation of the wetted surface and its evolution over time [70]. 

The two plots included in the study showed slightly different seasonal patterns. These would not necessarily be associated to the fact that the installed sensors corresponded to different models, since the two plots were managed separately and, at specific periods, received different irrigation doses. In Plot I, the sensors reported drier measurements, especially at 15 cm depth, than expected according to the HYDRUS-3D simulations. Neutron probe measurements supported the results of the simulations [37]. A possible explanation for this divergence could be the thinner spatial resolution of sensors, together with a scenario of limited irrigation doses. In Plot II, the general agreement with the model was better but nonetheless the sensors still differed in terms of the precise response to all water input/output disruptions. To check the effect of specific soil calibration, the 10HS capacitive sensors installed in Plot II were specifically calibrated for this soil at the end of the study period [43]. When the measured data were recalculated according to the specific soil calibration, the overall set of calibrated measurements centered around those of the simulations, but the same scatter persisted. Noticeably, measurements at some specific locations systematically increased or decreased when the soil-specific calibration was applied. Compared with the simulations, the sensors tended to give drier measurements at A60 and wetter ones at B30 and C30. This could be because the root water uptake reached higher depths than assumed in the simulations. 

### 4.3. Sensor Sensitivity at Each Location

Despite the complexity involved in discriminating between relevant trends and the noise introduced by variability between sensors, there is no doubt that important information for irrigation control can be derived from sensor response to irrigation cycles. These responses vary with sensor position and depth and must be taken into account both when designing sensor deployment and later on when interpreting the recorded data. Among other things, the effect of sensor position varies according to the soil hydraulic properties, meteorological conditions, and the irrigation configuration [71].

A feature of interest for the usage of soil moisture sensors is their ability to indicate the balance of water inputs and outputs to the soil. We observed that, depending on their location, they are more sensitive either to the recent balance of the last irrigation cycle or to the aggregated balance of several cycles. In particular, the sensors located in position A responded to the water balance of the same day and were especially more sensitive at the depths of 15 cm and 30 cm. Importantly, the sensors installed in these positions were not sensitive to the water balance of the previous week. This suggests that, at these locations, sensors tend to respond to irrigation rapidly and intensely, with little memory of the soil water trends of a few days earlier. The sensors installed in position B at all depths were sensitive to both the water balance of that day and to that of the previous week, especially at the 30 cm depth. The sensitivity at this location, B30, can be explained in terms of the progressive effect of the water balance over the course of several days on the SWC at this point through the overlapping or recession of two neighboring wet bulbs. The sensors located at C15 were sensitive to the balance of the previous week but not to that of the last day. Additionally, sensors located at the 30 cm depth and, to a lesser extent, sensors located at the 60 cm depth retained a memory of the water balance of that day and the previous week. This may be because the sensors in these positions were located on the perimeter of the wet area and were only affected by the dynamics of a single wet bulb.

Overall, this different sensitivity, either to the last cycle or to the aggregated period, is also observed in the simulations. The difference is that the B15 and B30 sensors seem more sensitive to the aggregated balance than the simulations. Compared to the simulations, some sensor locations (B60, C15, D30) are less sensitive to the last irrigation cycle and more sensitive to the aggregated balance. This may be due to the actual noise of the irrigation cycles at these positions, while the effect of the accumulated balance is more straightforward. 

### 4.4. Contributions of Different Factors to Sensor-To-Sensor Differences

Measurements of SWC by capacitance-type soil sensors show large differences between sensors. Factors such as the calibration applied, the effect of soil temperature and the variability in the area wetted by drippers can contribute individually and in an accumulative way to differences in the measured SWC values. Comparing the potential range of uncertainty by each of these contributory factors could provide clues as to the relative importance of each of them for ensuring the quality of the measurements. 

The calibration applied has a minor effect on the variability of the SWC measured by sensors. The potential effect of soil temperature, even if detectable in EC-5 sensors, was also limited. The measurements of the virtual HYDRUS-3D sensors can be used to explain a small part of the differences between real sensors. A large part of the impaired repeatability of the sensors may be attributable to the variability observed in the wetted area. In the HYDRUS-3D simulations, this wetted area variability is even larger than that of the sensor measurements, which might be explained by the trend, observed in our data, of the variability increasing as the distance to the soil surface decreases. The results suggest that most of the variability observed in the sensor measurements is caused by the arbitrariness of the shape and the positioning of the wet bulbs. In other words, the variability in a drip-irrigated orchard may be caused by the co-occurrence of a sharply defined nature of the actual distribution of soil water and a small volume perceived by each capacitance sensor.

Sensor performance in laboratory conditions suggests that their lack of repeatability in the field is not a fault of the sensors but a consequence of the actual complexity of the soil environment in drip-irrigated orchards. Capacitance sensors perceive a smaller soil volume than desirable to compensate for small-scale soil variability, as these sensors are too sensitive to local variations in soil texture, and the presence of gravel, stones, roots, macropores or small compacted soil parts [72]. In addition, wet bulbs may coexist with patches of differently shaded/sunlit soil spots. Roots may be more clustered and unpredictable than in annual crops. At the same time, macropores and differences in soil bulk density may develop more easily than in arable crops. 

### 4.5. Recommended Location for Capacitance Sensors in Drip Irrigation

Despite the accuracy of capacitance-type soil moisture sensors in laboratory conditions, in actual drip-irrigated orchards their usage is complicated by both their low repeatability and the dependence of their performance on their location in the soil. The non-uniform distribution patterns make soil water sensor placement a key factor in automated irrigation scheduling [73]. The plant root architecture around the drippers also complicates the decision as to where to place moisture sensors [74]. As a result, in drip-irrigated orchards, the approach when using these sensors cannot be the same as in scenarios of more homogeneous soil water distribution, such as in rainfed or sprinkler-irrigated field crops. In particular, any approach relying on an accurate assessment of the SWC or its projection to the volume of available water can be unreliable. Instead, an alternative is to use an approach which focuses more on the SWC trends by individual sensors [75]. 

The optimal location of capacitance-type moisture sensors for SWC monitoring depends on the type of crop, soil texture, salinity, and irrigation system, among other things. Various authors, including Soulis and Elmaloglou [71], have investigated the effects of sensor position and accuracy on drip irrigation scheduling. These same authors [76] introduced the time stable representative position (TSRP) concept and proposed general guidelines for sensor placement in soil moisture-based surface and subsurface drip irrigation scheduling systems (28 cm below the soil surface and 15 cm from the dripline). In a subsequent study [10], they complemented their previous work by considering the representativity of SWC readings and the TSRP in two layered soil profiles. They determined that optimum sensor positions for drip irrigation in a layered soil were at a horizontal distance of 7 cm and a depth of 16 cm in the upper layer, and at a horizontal distance of 11 cm and 34 cm depth in the lower layer.

Regarding the results of this study, attributes that are of interest for irrigation management include sensor-to-sensor repeatability, the extent to which sensor position represents the overall soil water availability to the crop, and the ability of the sensor location to match applied irrigation doses and actual irrigation needs. Our results indicate each of these attributes has its own pattern of response at different sensor locations, and that probably there is no single location that best reflects these attributes. Moreover, the optimal trade-off between these attributes may depend on the precise purpose and type of usage of the sensors in the farm in question. In this respect, the criteria for sensor deployment intended for visual supervision of soil water may prioritize obtaining a wide and clear view of the whole soil, whereas sensor deployment intended for automated irrigation scheduling may prioritize robustness and sensitivity to changes in the soil water budget. 

Nevertheless, when used for irrigation control, the criteria for selecting sensor locations would also depend on the control algorithm used. In this respect, if the control algorithm is based on thresholds for activating/deactivating irrigation pulses [15,77] the criteria may differ from when the algorithm is based on a water balance approach and tuned through sensor feedback [39,75]. Therefore, the optimal choice of sensor location will depend on the intended usage. Some authors [78], in a study on banana crops, established that the optimal position of the sensors for irrigation scheduling purposes varied according to the crop growth stage. Other authors [57,75,79] used two or more depths to monitor SWC. Alternatively, rather than combining different depths, for automated irrigation it makes sense to ensure measurement robustness by focusing on repetitive positions where there is more root activity [39,72]. Our study suggests that, to provide feedback to an irrigation scheduling algorithm based on water balance tuned by sensors [39], the combination of sensors close to the vertical of the dripper (location A30) with others in the mid-point location between two neighboring drippers (location B30) provide useful and complementary information. Moisture sensors aligned with the dripper provide an immediate response to the cycles of irrigation and water uptake by roots, while sensors between two drippers tend to display a slower dynamic which better represents the cumulative balance of the preceding period of several days. In our results, the best performing depth of 30 cm coincides with peak root activity. Other depths seem less favorable, with sensors at 15 cm being the least repeatable and sensors at 60 cm being the least responsive to the irrigation cycles.

Our proposals for capacitive sensor location for irrigation control are in line with other authors who used tensiometers on drip irrigated crops. Hodnett et al. [80] recommended installing tensiometers along the dripline, below the root zone, and inside the wet zone. Thompson et al. [81], who considered the subsurface drip-irrigation of broccoli, suggested placing tensiometers midway between two plants located in the same row at a depth of 30 cm. However, Dabach et al. [82], using HYDRUS 2D/3D to evaluate the optimum tensiometer location with ψ measurements in heterogeneous soil, determined that the optimal location was near the subsurface dripper. In addition, Nolz et al. [83], who monitored the soil water in a vineyard with Watermark sensors, determined that the representative measurement depth of water absorption by plants was 30 cm.

## 5. Conclusions

This study describes the variability in the measurements of SWC collected by capacitance-type soil moisture sensors in conditions of drip-irrigated orchards and analyses them in terms of uncertainty of the measurement process and possible variability of the actual quantities. The observed differences in sensor measurements were compared with the estimated potential perturbation as the result of factors such as the variability in the wetted area below the drippers, soil temperature, sensor calibration, and the gradients of SWC within a wet bulb expected by simulations. The results obtained show that moisture sensors installed in the field experience more variability than the simulations. Our results suggest that the main source of uncertainty involved in these measurements is the exact positioning of the sensor within the actual wet bulbs, as these vary in size, shape, and alignment with respect to the dripper in a magnitude that may explain the observed sensor-to-sensor differences. For its part, uncertainty in the measurements resulting from sensor calibration is only a fraction of the observed variability in data collected by sensors. This indicates that an increased accuracy in SWC measurements is considerably less relevant compared to the variability associated with the wetting pattern. The effect of temperature, with variation throughout the day and according to the position of the dripper, was especially notable in the EC-5 sensors. The soil water dynamics represented by the HYDRUS-3D simulation could only explain a small part of the differences observed in the real sensors. These simulations probably correspond to an ideal wet bulb, symmetric and centered around the dripper with homogeneous soil characteristics and root distribution, in contrast with the arbitrary and sharply defined variations in these conditions that can occur in actual wet bulbs.

According to the sensor response to irrigation, sensors closer to the dripper in position and depth (A15) respond quickly, have the highest amplitude and lowest repeatability, and are sensitive to the water balance of the same day. For their part, the sensors positioned at greater depth and further away from the dripper (C60) respond slightly, have the lowest amplitude and highest repeatability, and are sensitive to the water balance of the whole previous week. 

The analysis of the soil water dynamics allows the definition of candidate regions for monitoring. Positions and depths that provide more information for automated irrigation scheduling in a drip-irrigated orchard are also of interest to better understand the soil water dynamics. Given the variability of the system, it is convenient to locate sensors in repeat positions to make the interpretation more robust. In the context of automated irrigation scheduling based on the water balance tuned by soil moisture sensors, the recommended sensor locations could be a combination of sensors close to the vertical of the dripper (position A) and other sensors midway between neighboring drippers (position B), both at 30 cm depth.

## Figures and Tables

**Figure 1 sensors-20-05100-f001:**
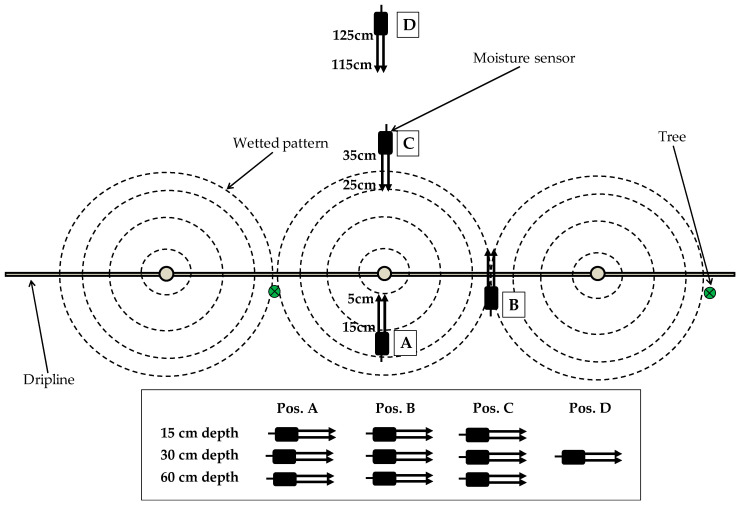
EC-5 and 10HS moisture sensors installed at three depths (15, 30 and 60 cm) in four positions relative to the dripper (Pos. A: center of wet bulb, Pos. B: mid-point between two drippers, Pos. C: perimeter of the wet area, Pos. D: outside the influence of the dripper.

**Figure 2 sensors-20-05100-f002:**
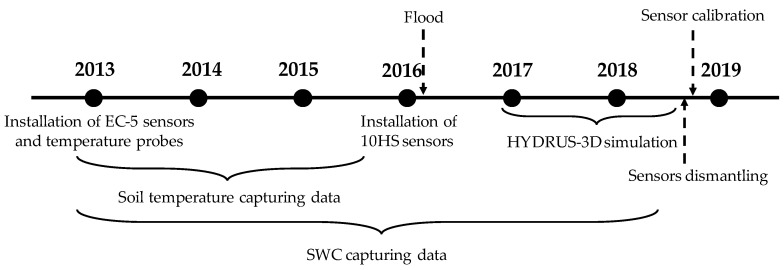
Temporal sequence where the most relevant moments are indicated. Recording of soil temperature ended in early 2016 when the setup was damaged by flood. The soil water content (SWC) measurements analyzed in this paper focus on the irrigation seasons of 2017 and 2018, which were also simulated with HYDRUS. At the end of this period, all 10HS sensors were dismantled and calibrated in laboratory.

**Figure 3 sensors-20-05100-f003:**
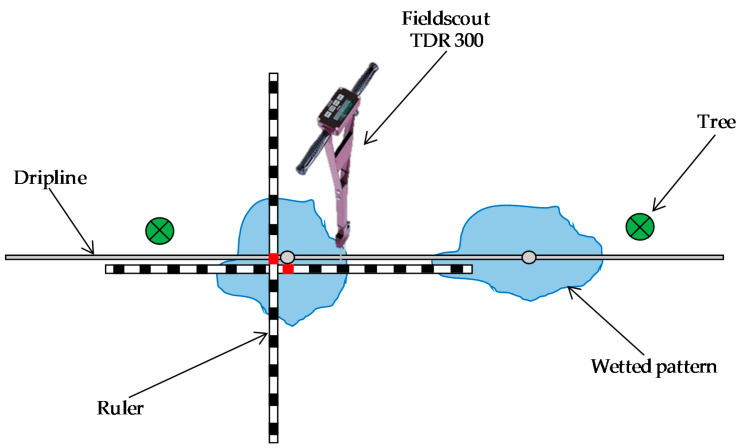
Characterization of the wetting patterns around drippers using the Fieldscout time domain reflectometry (TDR) 300.

**Figure 4 sensors-20-05100-f004:**
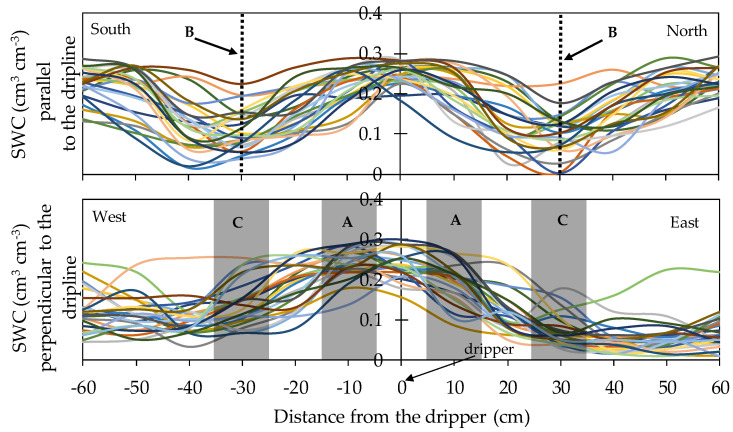
SWC measurements variability parallel and perpendicular to the dripline using the Fieldscout TDR 300. The different colors represent the repetitions.

**Figure 5 sensors-20-05100-f005:**
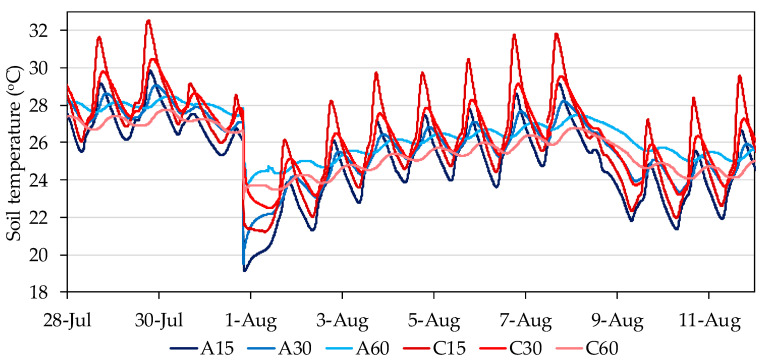
Sample of daily fluctuations in soil temperature at the studied positions and depths in Plot I in July–August 2015.

**Figure 6 sensors-20-05100-f006:**
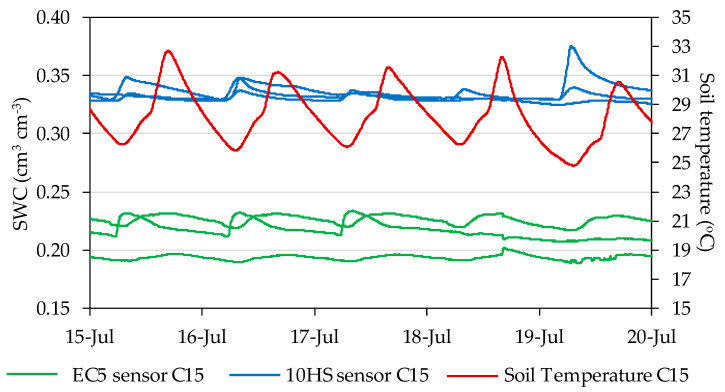
Comparison of daily fluctuations in soil temperature and SWC at position C, 15 cm depth in Plot I in July 2015.

**Figure 7 sensors-20-05100-f007:**
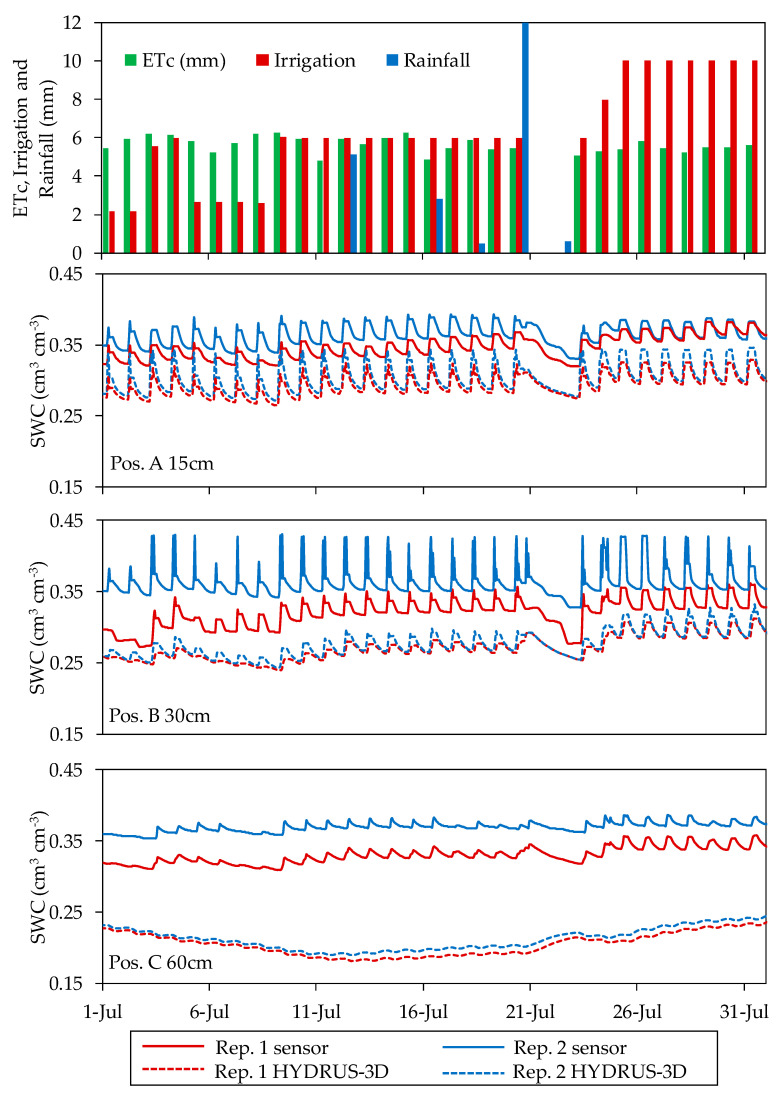
Example of daily behavior of humidity sensors located in position A at the depth of 15 cm, B at the depth of 30 cm, and C at the depth of 60 cm in Plot II in July 2018.

**Figure 8 sensors-20-05100-f008:**
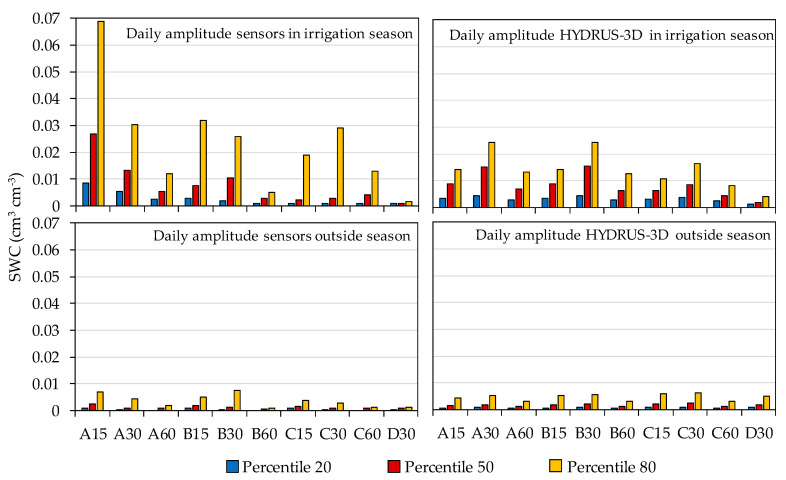
Daily amplitude of the measured and simulated soil water contents at different position and depth.

**Figure 9 sensors-20-05100-f009:**
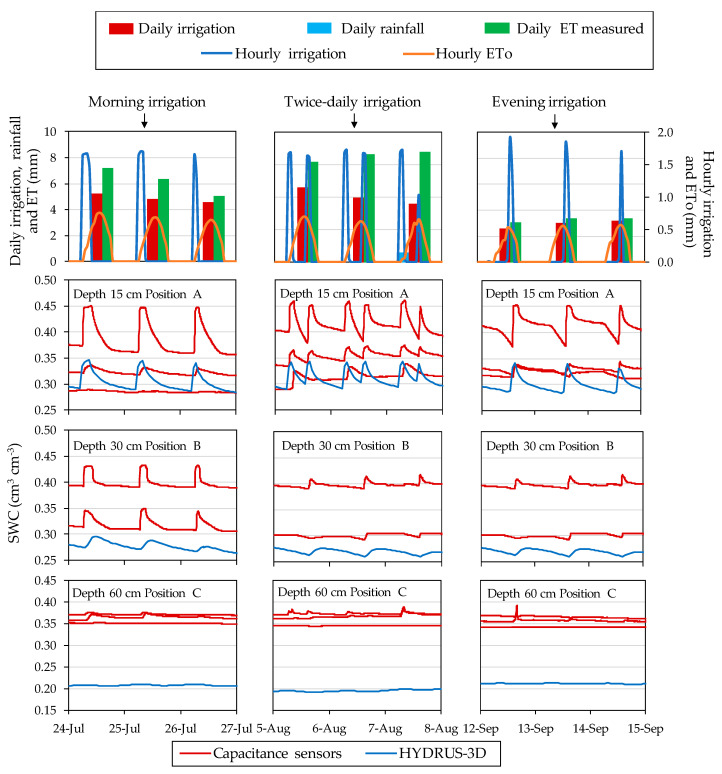
Relationship between the daily shape of the SWC curve and the moment of the day of irrigation in Plot II in 2017. In red are represented sensor moisture repetitions and in blue is represented the model.

**Figure 10 sensors-20-05100-f010:**
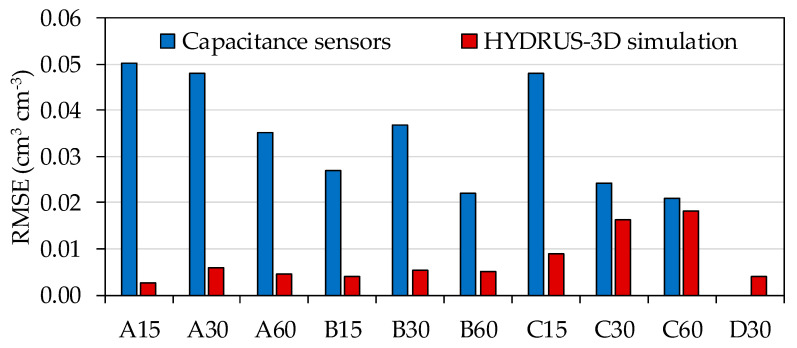
Sensor-to-sensor differences in SWC measured by capacitance sensors and by HYDRUS-3D simulations at the same soil positions ±10 cm in the direction to the dripper.

**Figure 11 sensors-20-05100-f011:**
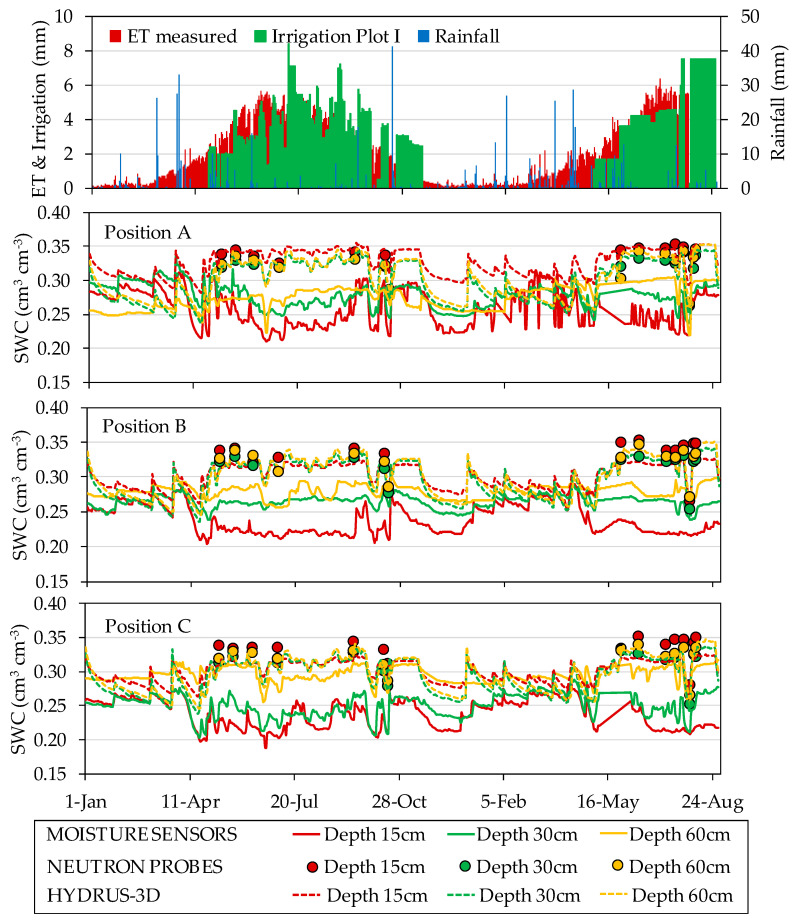
Seasonal variations in soil water content (represented by the series of daily minimum values, cm^3^ cm^−3^) measured by EC-5 capacitance soil moisture sensors, neutron probes and HYDRUS-3D simulations in different positions and depths in the years 2017 and 2018 in Plot I. Irrigation, rainfall, and ET measured by the weighing lysimeter is detailed at the top of the figure.

**Figure 12 sensors-20-05100-f012:**
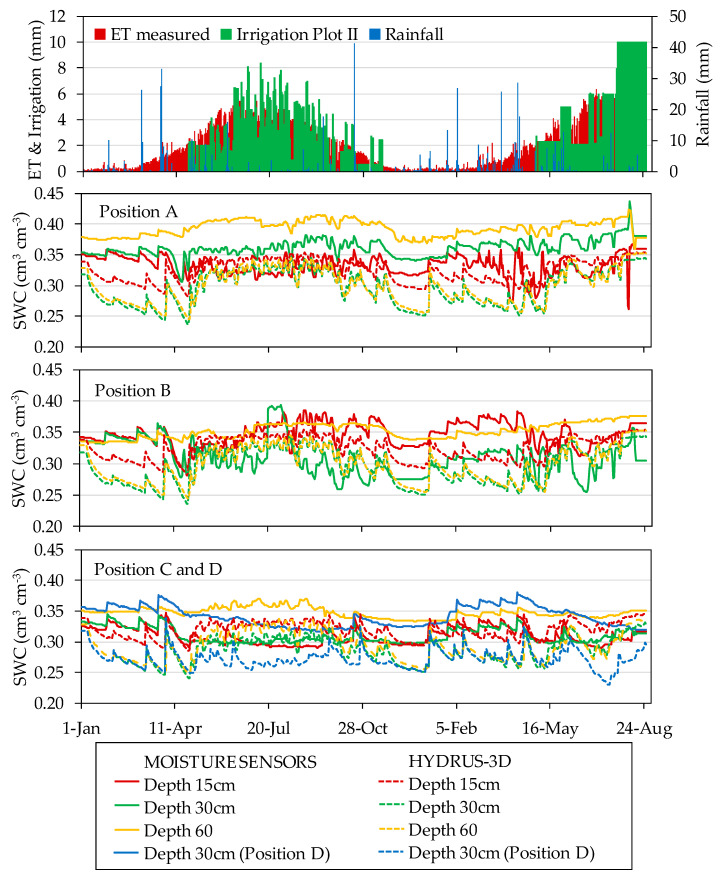
Seasonal soil water content (cm^3^ cm^−3^) measured by 10HS moisture sensors and HYDRUS-3D simulations in different positions and depths in the years 2017 and 2018 in Plot II. Irrigation, rainfall, and ET measured by the weighing lysimeter is detailed at the top of the figure.

**Figure 13 sensors-20-05100-f013:**
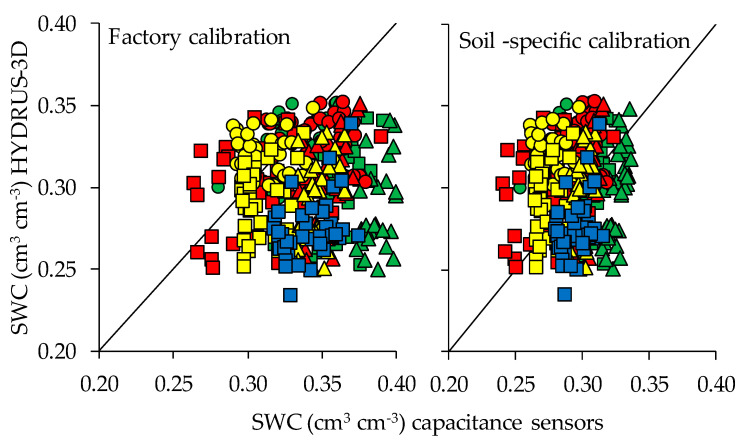
Fit between soil water content (SWC) measured by capacitance sensors and estimates by HYDRUS-3D, comparing factory calibration of the sensors with soil-specific calibration. Data includes one measurement every 15 days during the whole studied period. Colors indicate sensor position relative to dripper (green = A, red = B, yellow = C, blue = D) and shapes indicate depths (◯ = 15 cm, □ = 30 cm, ∆ = 60 cm).

**Figure 14 sensors-20-05100-f014:**
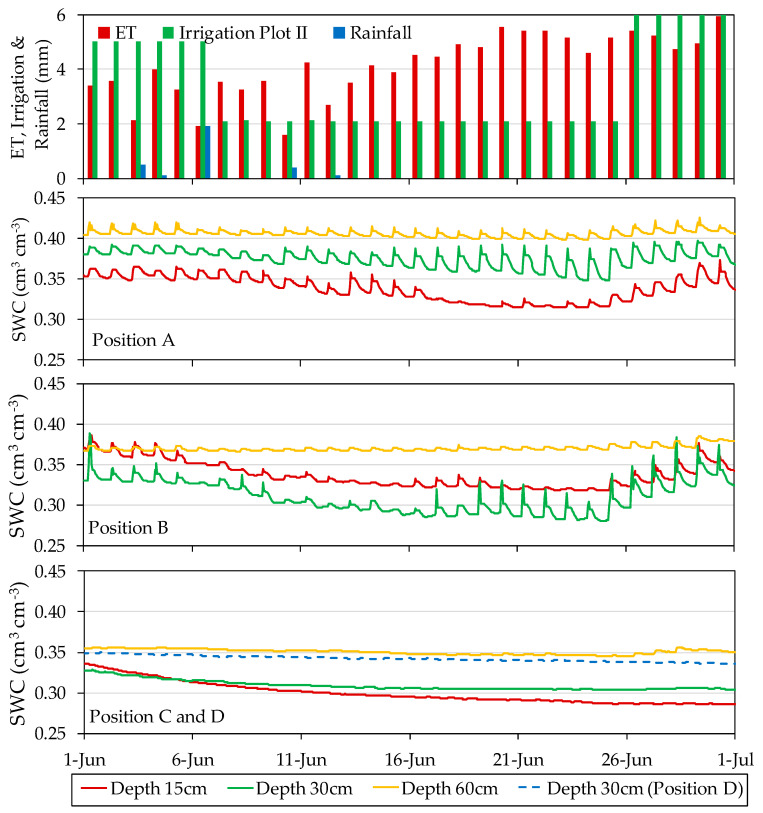
Sample of different types of sensor responses to the soil water balance, corresponding to different sensor locations in Plot II in June 2018.

**Figure 15 sensors-20-05100-f015:**
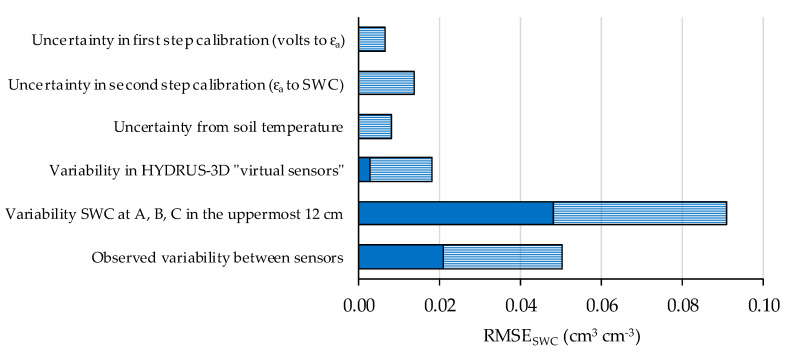
Ranges of uncertainty and variability in the measurement of soil water content (SWC) by capacitance sensors. Dark blue indicates least variability and light blue indicates maximum variability/uncertainty.

**Table 1 sensors-20-05100-t001:** Physical soil properties at three depths.

Depth (cm)	0–20	20–40	40–60
**Sand (%)**	35.80	35.50	36.00
**Silt (%)**	40.70	40.60	39.90
**Clay (%)**	23.50	23.90	24.10
**USDA soil classification**	loamy	loamy	loamy
**Bulk density (g cm^−3^)**	1.48	1.50	1.53
**Organic matter (%)**	1.99	1.57	1.34

**Table 2 sensors-20-05100-t002:** Deployment of 10HS sensors in the study plot.

Position	Distance to Dripper (cm)	Depth (cm)
A: center of wet bulb	5–15	15, 30 and 60
B: mid-point between two drippers	25–35	15, 30 and 60
C: wet area perimeter	25–35	15, 30 and 60
D: outside the influence of the dripper	115–125	30

**Table 3 sensors-20-05100-t003:** Summary of dependence of soil water content (SWC) measured by sensors on the measurement of a week before (Coef_SWC_7_), the water input/output balance of the day (Coef_bal) and on the water input/output balance of the whole previous week (Coef_bal_7_). N corresponds to the number of SWC measurements.

Position	Depth (cm)	N	R^2^_Adj	Coef_SWC_7_		Coef_bal		Coef_bal_7_	
A	15	514	0.994	1.0021	***	0.0036	***	0.0007	n.s
A	30	514	0.998	0.9968	***	0.0025	***	0.0000	n.s
A	60	514	0.999	0.9971	***	0.0009	**	0.0003	n.s
B	15	514	0.997	0.9876	***	0.0016	***	0.0021	***
B	30	514	0.982	0.9630	***	0.0039	***	0.0034	***
B	60	514	1.000	0.9973	***	0.0004	*	0.0013	***
C	15	514	0.998	0.9808	***	−0.0002	n.s	0.0034	***
C	30	514	0.998	0.9892	***	0.0018	***	0.0014	***
C	60	514	0.999	0.9963	***	0.0013	***	0.0009	**
D	30	260	1.000	0.9927	***	−0.0005	**	0.0013	***

n.s, *, **, *** are statistically non-significant, and statistically significant at *p* < 0.05, 0.01, or 0.001, respectively.

**Table 4 sensors-20-05100-t004:** Summary of dependence of soil water content (SWC) simulated by HYDRUS-3D on the simulations of a week before (Coef_SWC_7_), the water input/output balance of the day (Coef_bal) and on the water input/output balance of the whole previous week (Coef_bal_7_). N corresponds to the number of SWC measurements.

Position	Depth (cm)	N	R^2^_Adj	Coef_SWC_7_		Coef_bal		Coef_bal_7_	
A	15	514	0.999	0.9979	***	0.0020	***	0.0004	n.s
A	30	514	0.996	0.9963	***	0.0032	***	0.0006	n.s
A	60	514	0.997	0.9934	***	0.0023	***	0.0019	***
B	15	514	0.999	0.9974	***	0.0021	***	0.0003	n.s
B	30	514	0.996	0.9951	***	0.0031	***	0.0007	n.s
B	60	514	0.997	0.9929	***	0.0022	***	0.0021	***
C	15	514	0.999	0.9936	***	0.0017	***	0.0010	***
C	30	514	0.997	0.9892	***	0.0024	***	0.0019	***
C	60	514	0.998	0.9893	***	0.0016	***	0.0030	***
D	30	260	0.999	0.9756	***	−0.0007	***	0.0051	***

n.s, *** are statistically non-significant, and statistically significant at a *p* < 0.05, 0.01 or 0.001, respectively.

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
