# Peer review of "Analysis of the Variability in Soil Moisture Measurements by Capacitance Sensors in a Drip-Irrigated Orchard"

_sensors, 2020, doi:10.3390/s20185100_

Round 1

Reviewer 1 Report

    In this contribution the authors reported the variability in the measurements of SWC collected by capacitance-type soil moisture sensors in conditions of drip-irrigated orchards and analyses them in terms of uncertainty of the measurement process and possible variability of the actual quantities. This work is well thought out, explained clearly and important to the sensing community. Meanwhile, the paper is well written. There is only one small point that need to be addressed before the paper will be ready for publication.

1) A little more detailed information about soil moisture measurements is needed. In introduction, it is better to address the other competitive methods for measuring soil moisture and the pros and cons of capacitance-type moisture sensors. 

Author Response

A little more detailed information about soil moisture measurements is needed. In introduction, it is better to address the other competitive methods for measuring soil moisture and the pros and cons of capacitance-type moisture sensors.

Response: Ok, now the text reads: “Soil moisture can be measured using electromagnetic methods, such as time domain reflectometry (TDR) and capacitance sensors, or using electrical resistance blocks, neutron probes or tensiometers. Among the range of different soil water sensing technologies, capacitance-type soil moisture sensors are the most popular because of their cost, reasonable robustness and precision, low power consumption and low maintenance requirements”

Reviewer 2 Report

General comments:

This is a very interesting, well-written, and perhaps too long paper which raises the point of SWC measurements with electronic sensors (capacitive probes). Despite authors provide many explanations for their experimental work they should explain it more clearly as the huge amount of data is difficult to handle and properly acquired by the reader. Despite there are already many figures, a new figure or table explaining clearly the temporal sequence when the sensors were installed and measurements taken, is necessay. The reader doesn’t understand clearly why T was taken two years before the real SWC measurements with capacitive probes, and used to feed the model HYDRUS-3D which needs soil T measurements. In the same way authors provide Figures (Fig 1 and Fig 2) illustrating the position of installed sensors, readers would appreciate a figure or a table illustrating the temporal scheme for the different sensors (date of capturing data and use of the HYDRUS model, from 2013 to 2018, and even period of calibration after the experiment).

A sentence explaining that soil T from 2013-2015 can be extrapolated to the 2017-2018 period of SWC measurements and modelling would also be appreciated (despite it is well-known that soil T varies little on a seasonal basis, in an experimental work of this type where you get daily measurements, this explanations seems necessary).

Specific comments:

L94. The research with water content sensors was done in 2017 and 2018, but the temperature was measured in 2015 (L148-151). Please explain.

L136. Fig 1: If this figure if for both types of sensors (EC5, 10HS) this should be indicated clearly in the caption.

L162. Why “reference wetting pattern”. Please explain.

L170. Please, summarize in a short paragraph what is said in Dominguez-Niño et al, 2020.

L297-299. Explain how weather conditions in this particular site influence soil T and how this might influence HYDRUS model, especially if you compare with SWC measured two years later.

L320. Figure 5: please complete the legend of colored lines. Only the explanation of the blue line is provided.

L340. Figure 6: repeated “Jul-18” for the X axe is little indicative. Please indicate the exact day. Perhaps the 12 mm rainfall deserves some comment as no SWC peak is apreciated: runoff? Absorbance by surface mulch?

L343-345: Authors should explain in the Methods section the advantages and disadvantages of the simulation procedure with regards a real measurement procedure. Like it is now, the first method is given for granted in Results and in Fig 7. 

L359. Authors should explain that Hydrus use simulated T data (2013) and this fact might be related or not to the differences with real SWC.

L351-353. Indicate “data not shown” or is it Fig 7?

L373-388. Not clear what your results are in this long paragraph: capacitance probes from 2017-2018 and HYDRUS model with data from 2013-2015 (Fig 8)?

L391. Fig 8. The caption of this figure is not complete as there are ET data (can we say estimated as data correspond to previous week or really measured?)

L448. Fig 10. Similar comment as in Fig 8 and corresponding explanations in the text.

L455. Fig 11: similar comment as in Fig 8 and corresponding explanations in the text.

L506: Explain what is N in table 3 (number of SWC measurements?).

L525. Fig 13: Explain in self-understandable mode the caption of this figure. Please indicate with a discontinuous line the readings from position D. Like it is now it is quite confusing.

L592-595. Consider reviewing the English of this long sentence and better split in tow sentences as it contains two different concepts.

L845: Refer to what should be explained before regarding the fact that soil T and SWC were not measured at the same time.

Author Response

This is a very interesting, well-written, and perhaps too long paper which raises the point of SWC measurements with electronic sensors (capacitive probes). Despite authors provide many explanations for their experimental work they should explain it more clearly as the huge amount of data is difficult to handle and properly acquired by the reader. Despite there are already many figures, a new figure or table explaining clearly the temporal sequence when the sensors were installed and measurements taken, is necessary. The reader doesn’t understand clearly why T was taken two years before the real SWC measurements with capacitive probes and used to feed the model HYDRUS-3D which needs soil T measurements. In the same way authors provide Figures (Fig 1 and Fig 2) illustrating the position of installed sensors, readers would appreciate a figure or a table illustrating the temporal scheme for the different sensors (date of capturing data and use of the HYDRUS model, from 2013 to 2018, and even period of calibration after the experiment).

Response: Ok. Now the manuscript includes a temporal scheme indicating the most relevant moments related to the installation of soil moisture and temperature sensors, calibration and simulation with HYDRUS-3D.

Soil temperature probes and EC-5 SWC sensors were installed in 2013 and were recorded together between 2013 and early 2016. Then, the setup was damaged by an accidental flood and temperature measurements ended, while measurements of SWC continued. In this manuscript, the measurements of soil temperature were used only for checking if either their fluctuations or their gradients in the soil might explain the variability in SWC recorded by capacitance soil moisture sensors. We considered that data from simultaneous measurements of soil temperature and SWC in 2015 were sufficient for showing that, even though the effect of temperature could be noticed in sensors EC-5, the magnitude of this effect was too small to account as the main source of variability for the SWC measurements. According to the manufacturer, sensors 10HS had an improved stability in front of temperature fluctuations compared with sensors EC-5. In our data, this improvement could be noticed as a flat response of measurements by 10HS in periods without water inputs/outputs though with expected fluctuations in soil temperature. The magnitude of the perturbation of soil moisture measurement by temperature measured in EC-5 sensors was also taken as an estimate of the potential perturbation on 10HS sensors, though in 10HS the perturbation should be smaller.

The analysis of SWC data conducted in this manuscript focuses on the period 2017-2018 because this is the period simulated with HYDRUS. It is worth to note that all sensors had been installed at least one year before the analyzed period and, hence, their surrounding soil had not been recently perturbated by the installation process.

The HYDRUS model does not use soil temperature as an input. It does use transpiration, which could be labelled “T” in other documents and this coincidence may cause some confusion. In our work, the transpiration data used by HYDRUS was measured by the weighing lysimeters in the simulated period (2017-2018).

Now the text reads (lines 151-160): “A total of nine temperature probes (Omega HSTH-44000, 2252 Ohm) were also installed in 2013 in Plot I, at soil locations equivalent to those of the EC-5 sensors. Probe readings were recorded until the end of 2015 using the same datalogger as for the EC-5 sensors, through a dedicated multiplexer AM16/32 (Campbell Scientific Inc., Logan, UT, USA). Soil temperature measurements ended in early 2016 when the multiplexer was damaged by a flood. The dataset of soil temperatures analysed in this work corresponds to the 2015 season, when soil temperature and soil moisture were recorded simultaneously. Figure 2 shows a temporal scheme that indicates the most relevant moments related to the installation of soil moisture and temperature sensors, calibration and simulation with HYDRUS-3D.”

A sentence explaining that soil T from 2013-2015 can be extrapolated to the 2017-2018 period of SWC measurements and modelling would also be appreciated (despite it is well-known that soil T varies little on a seasonal basis, in an experimental work of this type where you get daily measurements, this explanations seems necessary).

Response: Ok. Now the text reads (lines 319-323): “Soil temperature measurements ended accidentally in early 2016. We considered that the simultaneous recording of soil temperature and soil moisture by EC-5 probes in 2015 was sufficient to assess the magnitude of the effect of temperature on the measurements of soil moisture and that this magnitude was also representative for 2017 and 2018.”

  • Line 94: The research with water content sensors was done in 2017 and 2018, but the temperature was measured in 2015 (L148-151). Please explain.

Response: As commented above, the newly added Figure 2 depicts a temporal sequence with the periods of measurement and other activities. The only use of soil temperature in this manuscript was to assess the magnitude of potential perturbation by soil temperature on the measurements of SWC. The dataset of 2015, which included simultaneous measurement of soil temperature and SWC by EC-5 sensors, was used for determining the effect of temperature on the measurements of soil moisture. The magnitude of this effect was determined with data from 2015 and assumed representative of other years at the same site, since neither soil properties nor climate conditions changed substantially. On the other hand, the main body of the analyses of SWC variability deals with the irrigation seasons of 2017 and 2018 because this was the period when sensor data could be compared with simulations and with measurements by neutron probe.

Neither processing of sensor data in the period 2017-2018 nor simulations with HYDRUS required data of soil temperature.

  • Line 136: Fig 1: If this figure if for both types of sensors (EC5, 10HS) this should be indicated clearly in the caption.

Response: Ok. Now the caption reads “EC-5 and 10HS moisture sensors installed at three depths (15, 30 and 60 cm) in four positions relative to the dripper (Pos. A: centre of wet bulb, Pos. B: mid-point between two drippers, Pos. C: perimeter of the wet area, Pos. D: outside the influence of the dripper”.

  • Line 162: Why “reference wetting pattern”. Please explain.

Response: It was considered “reference wetting pattern” because was the wetting pattern most frequently observed in the whole plot. We need a “reference wetting pattern” to compare each individual wetting pattern. Then, we choose the most frequent wetting pattern observed in that plot.  

  • Line 170: Please, summarize in a short paragraph what is said in Dominguez-Niño et al, 2020.

Response: Now the paragraph is more completed, and it reads “Twelve access tubes were installed in 2013 at positions A, B, C, D, relative to the drippers, and repeated in three drippers as described in Domínguez-Niño et al., 2020. The volumetric soil water content in these access tubes was measured using a neutron probe (Hydroprobe 503DR, Campbell Pacific Nuclear Corp., Martinez, CA, USA) which had previously been calibrated for this site., Measurements were taken at depths of between 0.20 m and 1.00 m at intervals of 20 cm depth, on a total of 15 days in the periods from May to October of 2017 and 2018”.

  • Line 297-299: Explain how weather conditions in this particular site influence soil T and how this might influence HYDRUS model, especially if you compare with SWC measured two years later.

Response: The daily pattern of soil temperature is mainly affected by the solar radiation reaching the soil surface that day. Hence, daily fluctuations of soil temperature are wider in sunny days than in cloudy days. In addition, rainfall used to cool abruptly the soil, especially at shallow soil depths. All these effects will determine that every single day would have its particular curve of soil temperature.

Our analysis did not look for the precise soil temperature at any given day or hour. Instead, it looked for the magnitude of potential perturbation on SWC measurements. To do so, we used the widest observed amplitude of the daily fluctuations and at the largest observed differences in temperature between different soil locations. At a given site, these data should not change substantially between years.

The manuscript concludes that those ranges of fluctuations and differences between locations in soil temperature are too small to explain the much larger variability in SWC measurements by capacitive soil moisture sensors.

HYDRUS model does not use soil temperature as an input. All inputs of HYDRUS model were obtained from data of the simulated period (2017-2018).

  • Line 320: Figure 5: please complete the legend of colored lines. Only the explanation of the blue line is provided.

Response: Now the figure is corrected.

  • Line 340: Figure 6: repeated “Jul-18” for the X axe is little indicative. Please indicate the exact day. Perhaps the 12 mm rainfall deserves some comment as no SWC peak is apreciated: runoff? Absorbance by surface mulch?

Response: Now the figure it is corrected and indicate the exact day. Regarding the 12 mm rainfall, the corresponding peaks of SWC can be seen in Figure 6 as a secondary peak that day (after the peak of irrigation), especially at 15 cm and 30 cm depth. At 15 cm depth it can be observed the last peak before the fall of the curve. At 30 cm, it is clearly seen in the sensor represented as a blue line.

It must be noted that rainfall is represented in a diagram showing three bars per day and that the position of the rainfall bar is not aligned with the hour of the day when the rainfall occurred. This is why the bar of rainfall is not exactly aligned with the rise of SWC seen on the graphics below.

  • Line 343-345: Authors should explain in the Methods section the advantages and disadvantages of the simulation procedure with regards a real measurement procedure. Like it is now, the first method is given for granted in Results and in Fig 7. 

Response:

Now the text reads (lines 233-237): “HYDRUS-3D model is characterized by simulating soil water dynamics in a homogeneous soil and root distribution where ideal, symmetric and centred wet bulbs develop around the dripper. However, HYDRUS-3D neither represent heterogeneous soil and root distributions, or macropores and soil irregularities among other phenomena that usually take place in a real soil where wet bulbs are generated.”

  • Line 359: Authors should explain that Hydrus use simulated T data (2013) and this fact might be related or not to the differences with real SWC.

Response: HYDRUS did not use simulated T data because HYDRUS does not use soil temperature as an input. It does use Transpiration, which is also labelled “T” in some texts. These simulations used Transpiration data actually measured by the weighing lysimeters during the simulated period (2017-2018).

  • Line 351-353: Indicate “data not shown” or is it Fig 7?

Response: Corrected. Now the text reads: “the HYDRUS-3D simulations showed in Figure 8 a similar order of amplitude to that of the sensors for locations A30, A60, B30”

  • Line 373-388: Not clear what your results are in this long paragraph: capacitance probes from 2017-2018 and HYDRUS model with data from 2013-2015 (Fig 8)?

Response: Completed. Now the text reads: “Figure 9 illustrates the dynamics of SWC when irrigation was in the morning, split between morning and afternoon, and in the afternoon, showing the sensor-recorded data at different locations in the soil together with the corresponding HYDRUS-3D simulation in the 2017 season”

  • Line 391: Fig 8. The caption of this figure is not complete as there are ET data (can we say estimated as data correspond to previous week or really measured?)

Response: ET data were measured on the plot since a weighing lysimeter was available on the farm. Now in the caption appears “Daily ET measured”

  • Line 448: Fig 10. Similar comment as in Fig 8 and corresponding explanations in the text.

Response: Completed in the text and captions. Now the text reads: “Figures 11 and 12 show the seasonal soil water dynamics at the different sensor locations measured with capacitance sensors and simulated HYDRUS-3D model, irrigation, rainfall and ET measured by the weighing lysimeter in the years 2017 and 2018”

  • Line 455: Fig 11: similar comment as in Fig 8 and corresponding explanations in the text.

Response: Completed in the text and captions. Now the text reads: “Figures 11 and 12 show the seasonal soil water dynamics at the different sensor locations measured with capacitance sensors and simulated HYDRUS-3D model, irrigation, rainfall and ET measured by the weighing lysimeter in the years 2017 and 2018”

  • Line 506: Explain what is N in table 3 (number of SWC measurements?).

Response: Corrected. Now the text reads “N corresponds to the number of SWC measurements”.

  • Line 525. Fig 13: Explain in self-understandable mode the caption of this figure. Please indicate with a discontinuous line the readings from position D. Like it is now it is quite confusing.

Response: Corrected the figure and caption. Now the text reads “Different type of sensor response to the soil water balance in different locations in Plot II in June 2018”.

  • Line 592-595. Consider reviewing the English of this long sentence and better split in two sentences as it contains two different concepts.

Response: Corrected and split in two sentences. Now the text reads “The performance of capacitance moisture sensors (EC-5 and 10HS) installed at different depths and positions relative to the dripper were evaluated for two years. In addition, it was evaluated the particular conditions of disturbance and the natural variability of soil water patterns in drip irrigation”.

  • Line 845: Refer to what should be explained before regarding the fact that soil T and SWC were not measured at the same time.

Response: The indicated sentence remains correct and unchanged. As clarified now by Figure 2 (see also Figure 6), regarding sensors EC-5, soil T and SWC were measured at the same time in 2015. In this manuscript, the measurements of soil temperature were used solely for checking it either their fluctuations or their gradients in the soil might explain the variability in SWC recorded by capacitance soil moisture sensors. Simultaneous measurements of soil temperature and SWC in 2015 were sufficient for discarding soil temperature as the main source of variability for the SWC measurements.

Reviewer 3 Report

The paper presents the results of the variability involved in the measurement of soil moisture by capacitance sensors in a drip-irrigated orchard in two seasons 2018 and 2019, preceded by correctly selected and described literature and concluded with a detailed discussion of the results.

Selected parameters of sensors like calibration, soil thermal conditions, wetting patterns, sensors locations (the dept and position) were studied in detail. The aim of the study was to find the most optimal location for capacitive sensors to effectively and accurately manage irrigation.

The scientific level of the publication is appropriate. The research was conducted and described correctly. The Authors achieved the set goals, however, there are some inaccuracies in the manuscript necessary to explain. In my opinion, the paper can be published in the Sensors journal after some revision.

L.14: I suggest to shorten the abstract, in my opinion it is too long in its current form..

L.112: there is: fertirrigation, why not just fertigation?

L.150: I’m not sure if I understood it right – the soil temperature were measured in seasons 2013-2015. Please the Authors to explain, why is the variability of soil moisture explained by soil temperature changes in the earlier 2015 season? Why was the soil temperature not monitored during the experiment in 2018 and 2019?

L.231: The Equation 7 consists of 3 variables, and the following is a description of the 8 variables. Please the Authors to sort it out.

L.256: The coefficient of determination (R2) and the RMSE are commonly used for the statistical analysis, therefore I do not think, there is a need to explain what they are and how to calculate them.

L.277: The Authors mention: ‘ centring of the wet bulb was displaced westwards by 13.6 ± 7.5 cm from the emitter, towards the centre of the tree line.’ OK, but why? How about the slope of the terrain? I did not find any information concerning the soil surface in the plots in the orchards. Was it perfectly flat or not? If not, there should be added some comments about it.

L.320: Figure 5 - the colored lines for EC5 sensor and soil temperature in the description below the figure are missing.

L.339: Figure 6 – what is the time resolution in the X axis?

L.408: There is:’ In particular, the depths of 15 cm and 30 cm showed greater repeatability (0.022 cm3 cm-3 and 0.026 cm3 cm-3) than the depth of 60 cm (0.037 cm3 cm-3).’ While in the Figure 9 greater repeatability is shown by B15 and B60, while less by B30.

L.418: Why there were such great differences between RMSE of the sensors and RMSE simulated?

In some places an extra space between the word and the punctuation mark is necessary.

Author Response

  • Line14: I suggest to shorten the abstract, in my opinion it is too long in its current form.

Response: Now the text reads: “Among the diverse techniques for monitoring soil moisture, capacitance-type soil moisture sensors are popular because of their low cost, low maintenance requirements and acceptable performance. However, although in laboratory conditions the accuracy of these sensors is good, when installed in the field they tend to show large sensor-to-sensor differences, especially under drip irrigation. It makes difficult to decide in which positions the sensors are installed and the interpretation of the recorded data. The aim of this paper is to study the variability involved in the measurement of soil moisture by capacitance sensors in a drip-irrigated orchard and, using this information, find ways to optimize their usage to manage irrigation. For this purpose, the study examines the uncertainties in the measurement process plus the natural variability in the actual soil water dynamics. Measurements were collected by 57 sensors, located at 10 combinations of depth and position relative to the dripper. Our results showed large sensor-to-sensor differences, even when installed at equivalent depth and coordinates relative to the drippers. In contrast, differences among virtual sensors simulated using a HYDRUS-3D model at those soil locations were one order of magnitude smaller. Our results highlight as a possible cause for the sensor-to-sensor differences in the measurements by capacitance sensors the natural variability in size, shape, and centring of the wet area below the drippers, combined with the sharply defined variation in water content at the soil scale perceived by the sensors”.

  • Line 112: there is: fertirrigation, why not just fertigation?

Response: Corrected. Now in the text appears fertigation.

  • Line 150: I’m not sure if I understood it right – the soil temperature were measured in seasons 2013-2015. Please the Authors to explain, why is the variability of soil moisture explained by soil temperature changes in the earlier 2015 season? Why was the soil temperature not monitored during the experiment in 2018 and 2019?

Response:  Figure 2 has been added to clarify the temporal sequence of the diverse measurements and simulations. Temperature probes were installed, together with sensors EC-5, in 2013. However, the setup for measuring soil temperature was damaged by a flood in the winter of 2016. This is the reason why, in this manuscript, comparison of simultaneous measurement of soil temperature and SWC was done with data from 2015. The range of potential perturbation of SWC by soil temperature, characterized in 2015, should not change in the following years. On the other hand, most of the discussion on the variability of SWC in this manuscript focuses on the period 2017-2018 because that is the period simulated with HYDRUS.

  • Line 231: The Equation 7 consists of 3 variables, and the following is a description of the 8 variables. Please the Authors to sort it out.

Response: The equation 7 consist in different parameters, which are described below. Now the parameters have been ordered according to their order of appearance in the equation to improve their understanding.

  • Line 256: The coefficient of determination (R2) and the RMSE are commonly used for the statistical analysis, therefore I do not think, there is a need to explain what they are and how to calculate them.

Response: Despite the common use of R2 and RMSE in statistical analysis, the authors consider their explanation to understand the results obtained.

  • Line 277: The Authors mention: ‘centring of the wet bulb was displaced westwards by 13.6 ± 7.5 cm from the emitter, towards the centre of the tree line.’ OK, but why? How about the slope of the terrain? I did not find any information concerning the soil surface in the plots in the orchards. Was it perfectly flat or not? If not, there should be added some comments about it.

Response: At the scale of the overall plot the ground was flat, with no predominant slope on any direction. Nevertheless, at a centimetric scale, a pattern of microrelief in the soil surface near the dripper would condition the path of water before infiltration. The microrelief was in the range of 2-3 cm (from “valleys” to “hills”). The average shift of the wet area would suggest a predominant slope in the microrelief between the pipeline and the center of the tree line. However, this orientation of the microrelief pattern was not evident by visual inspection. In addition, the centering of the wet area had a large variability, which might be explained by the natural variability in these microrelief patterns.

  • Line 320: Figure 5 - the colored lines for EC5 sensor and soil temperature in the description below the figure are missing.

Response: Now the figure is corrected.

  • Line 339: Figure 6 – what is the time resolution in the X axis?

Response: Now the figure is corrected and indicate the exact day.

  • Line 408: There is: ’In particular, the depths of 15 cm and 30 cm showed greater repeatability (0.022 cm3 cm-3 and 0.026 cm3 cm-3) than the depth of 60 cm (0.037 cm3 cm-3).’ While in the Figure 9 greater repeatability is shown by B15 and B60, while less by B30.

Response: The data of the figure are the correct. Now the text is corrected according to the figure and reads:” In particular, the depths of 15 cm and 30 cm showed greater repeatability (0.027 cm3 cm-3 and 0.037 cm3 cm-3) than the depth of 60 cm (0.022 cm3 cm-3).

  • Line 418: Why there were such great differences between RMSE of the sensors and RMSE simulated?

Response: Our interpretation is that HYDRUS model only explains a small part of the differences observed between capacitance sensors. The HYDRUS-3D simulations correspond to an ideal wet bulb, symmetric and centred around the dripper with homogeneous soil characteristics and root distribution, in contrast with the arbitrary and sharply defined variations in these conditions that can occur in the actual wet bulbs encountered by sensors.

Round 2

Reviewer 3 Report

After I had read the Authors' explanations and the went through the revised article, I can conclude that several important corrections and clarifications have been made by the Authors, which significantly improved the scientific and utilitarian value of the manuscript.
In particular, the explanation regarding soil temperature and moisture measurements and additional Figure 2 clarified and improved the section of Materials and Methods. Suggestions for editorial corrections were also introduced.
Due to the above, I believe that the work can be published in the journal of  MDPI Sensors in it's current version.